# Dipole Pattern of Summer Ozone Pollution in the east of China and Its Connection with Climate Variability

Xiaoqing Ma[1], Zhicong Yin [123]

[1]Key Laboratory of Meteorological Disaster, Ministry of Education / Joint International Research Laboratory of Climate and Environment Change (ILCEC) / Collaborative Innovation Center on Forecast and Evaluation of Meteorological Disasters (CIC-FEMD), Nanjing University of Information Science & Technology, Nanjing 210044, China

[2]Southern Marine Science and Engineering Guangdong Laboratory (Zhuhai), Zhuhai, China

[3]Nansen-Zhu International Research Centre, Institute of Atmospheric Physics, Chinese Academy of Sciences, Beijing, China

**Corresponding author:** Zhicong Yin(yinzhc@nuist.edu.cn)

**Address:** No. 219 Ningliu Road, Pukou District, Nanjing University of Information Science & Technology, Nanjing 210044, China

**Tel.:** (+86) 136 5516 1661

**Abstract.**

Surface $O_3$ pollution has become one of the most severe air pollution problems in China, which makes it of practical importance to understand $O_3$ variability. A south-north dipole pattern of summer-mean $O_3$ concentration in the east of China (DP-$O_3$), which were centered at North China (NC) and the Pearl River Delta (PRD) respectively, has been identified from the simulation of a global 3-D chemical transport model for the period 1980–2019. Large-scale anticyclonic (cyclonic) and cyclonic (anticyclonic) anomalies over NC and the PRD resulted in a sharp contrast of meteorological conditions between the above two regions. The enhanced (restrained) photochemistry in NC and restrained (enhanced) $O_3$ production in the PRD contributed to the DP-$O_3$. Decreased sea ice anomalies near the Franz Josef Land and associated warm sea surface in May enhanced the Rossby-wave source over northern Europe and West Siberia, which eventually induced an anomalous Eurasia-like pattern to influence the formation of the DP-$O_3$. The thermodynamic signals of the southern Indian Ocean dipole were stored in the subsurface and influenced spatial pattern of $O_3$ pollution in the east of China mainly through the Hadley circulation. The physical mechanisms behind the modulation of the atmospheric circulations and related DP-$O_3$ by these two climate anomalies at different latitudes were evidently verified by large-scale ensemble simulations of the earth system model.

**Key words:** ozone pollution; sea ice; Eurasia pattern; sea surface temperature; meridional circulation

## 1. Introduction

Surface $O_3$ is an important air pollutant. Exposure to high concentrations of $O_3$ is detrimental to both human health and vegetation ecology (Rider and Carlsten, 2019). Since 2013, surface $O_3$ concentration has increased over most parts of China, which is largely attributed to changes in anthropogenic emissions (Xu et al. 2018). However, previous studies have shown that in addition to its trend of change, surface $O_3$ concentration also demonstrated large interannual variations with significant regional differences (Zhou et al. 2013; Chen et al. 2019). Based on analysis of 11 years of observational data over Hong Kong, Zhou et al. (2013) reported that the interannual variation of $O_3$ concentration observed during 2000–2010 could reach up to 30% of the annual average concentration. The $O_3$ concentration in Beijing also showed evident interannual variation during 2006–2016. For example, the $O_3$ concentrations in the summers of 2012–2013 were lower by about 10 ppbv than that in 2011 and 2014 (Chen et al. 2019).

High $O_3$ events are usually associated with meteorological factors (e.g., intense solar radiation, high air temperature and low humidity) favorable for $O_3$ formation, which can accelerate photochemical reaction and weaken the dispersions and depositions (Han et al. 2020). For example, Lu et al. (2019) designed sensitivity simulations to confirm that ozone pollution in China in 2017 was more serious than that in 2016, which was attributed to the large enhancement of nature emissions of ozone precursors caused by hot and dry climate condition in 2017. In the summer of 2013, the Yangtze River Delta experienced a severe heat wave with more stagnant meteorological conditions. The upper-level anticyclonic circulation with sink airflows led to abnormally low atmospheric water vapor content above the Yangtze River Delta and thus less than normal cloud cover, which was conductive to a strong solar radiation environment and significant increases in surface ozone (Pu et al. 2017). On the interannual to decadal time scale, anticyclonic anomalies over North China (NC) were critical for $O_3$ distribution in the summer and remotely linked with the effects of Eurasia teleconnection (EU) and west Pacific patterns (Yin et al. 2019).

The Arctic sea ice (SI) declined rapidly while its variability has been increasing over the past decades, which significantly affected summer atmospheric circulations over Eurasia (Lin and Li 2018). The preceding Arctic SI anomalies could aggravate anomalously high air temperature and drought disasters in NC by triggering EU-like atmospheric responses in summer (Wang and He 2015). Spring SI anomalies in the Barents Sea could prompt the Silk Road Pattern and resulted in a north-south dipole pattern of summer air temperature anomalies in the east of China (Li et al. 2021). When greater than normal SI occurred in the Barents Sea, local 500 hPa geopotential height would decrease and a wave-chain would form, which subsequently induced more precipitation in the south of East

China but less precipitation in the north (Wang and Guo 2004). Sea surface temperature (SST) in the Pacific and
Indian oceans also have significant effects on atmospheric circulation over the east of China (Li and Xiao 2021;
Xia et al. 2021). SST anomalies in the South China Sea and the equatorial Eastern Indian Ocean could trigger the
East Asian - Pacific pattern and resulted in a dipole pattern of summer temperature and precipitation in the east
of China, i.e., areas to the north of the Yangtze River became cold and wet, while areas to the south were hot and
dry (Han and Zhang 2009; Li et al. 2018). Tian and Fan (2019) found that winter SST in the southern Indian
Ocean might affect spring-summer SST anomalies near Australia. In summer, the anomalous Hadley circulation
in the western North Pacific played an important role in summer precipitation over the middle and lower reaches
of the Yangtze River.

71        Although great attention in previous studies has been paid to the increase of ozone pollution, little is known

about changes in the spatial pattern of summer-mean $O_3$ in the east of China. As revealed by Yin and Ma (2020),
the dominant pattern of daily-varying ozone pollution in the east of China showed an interannual variation that
was mainly driven by the large-scale western Pacific subtropical high and the East Asian deep trough. For example,
the frequent movements of the western Pacific subtropical high and the East Asian deep trough both contributed
to the out-of-phase variations in $O_3$ over North China and the Yangtze River Delta (Zhao and Wang 2017; Yin
and Ma 2020). However, to the best of our knowledge, whether the north-south dipole pattern of the summer mean
$O_3$ pollution existed in the east of China still remains unclear. In this study, we attempted to explore the dominant
pattern of summertime $O_3$ in the east of China and associated physical mechanisms behind. Its connections with
preceding climate variability were also examined. The remainder of this paper was organized as follows. The data
and methods are described in Section 2. Section 3 examined the dipole pattern of summertime $O_3$ in the east of
China and its possible influencing factors. The associated physical mechanisms were studied in Section 4. Major
conclusions and discussion are provided in Section 5.
**2.    Datasets and methods**
**2.1 Observations and Reanalysis Dataset**

86        Hourly ozone concentration observations from 2015 to 2019 were publicly available at

https://quotsoft.net/air/ and the last accessible data were for 23 September 2020. The relevant data were detrended
before all computations were conducted for the study period.

89        The meteorological fields data with a horizontal resolution of 0.5° latitude by 0.625° longitude for the period

1980–2019 were taken from the MERRA-2 dataset (Gelaro et al., 2017), including geopotential height at 500 hPa

(Z500), surface incoming shortwave flux (Ssr), low and medium cloud cover (Mlcc), precipitation (Prec), 10-m zonal and meridional winds (UV10m), and surface air temperature (SAT) and zonal and meridional winds and vertical velocity at different vertical levels. Monthly Outgoing Longwave Radiation (OLR) data ($1° \times 1°$) could be acquired from the University of Maryland OLR Climate Data Record portal (http://olr.umd.edu/). Monthly SI concentrations and SST ($1° \times 1°$) for the period 1980 - 2019 were downloaded from the website of the Met Office Hadley Centre (Rayner et al. 2003). Monthly mean subsurface ocean temperatures in the upper 250 m with a horizontal resolution of $1° \times 1°$ were obtained from the Met Office Hadley Centre EN4 version 2.1 (Good et al. 2013).

The wave activity flux (WAF) was computed to illustrate the propagation of Rossby wave activities (Takaya and Nakamura 2001):

$$W = \frac{1}{2|\bar{U}|} \begin{bmatrix} \bar{u}(\psi'^2_x - \psi'\psi'_{xx}) + \bar{v}(\psi'_x\psi'_y - \psi'\psi'_{xy}) \\ \bar{u}(\psi'_x\psi'_y - \psi'\psi'_{xy}) + \bar{v}(\psi'^2_y - \psi'\psi'_{yy}) \end{bmatrix}$$

where subscripts denote partial derivatives; the overbar and prime represent the climatological mean and anomaly, respectively; $\psi'$ represents the stream function anomaly. U is the horizontal wind speed; $u$ and $v$ are the zonal and meridional wind components, respectively; and $W$ denotes the two-dimensional Rossby WAF. The Rossby wave source $-\nabla \cdot V_x(f + \xi)$ proposed by Sardeshmukh and Hoskins (1988) is also calculated in this study. V, $\xi$ and $f$ refer to the horizontal wind velocity, relative vorticity and geostrophic parameter, respectively. $\nabla$ is horizontal gradient; subscript $\chi$ represents divergent component.

**2.2 1980–2019 O$_3$ concentrations simulated by GEOS-Chem**

Hourly ozone concentrations were simulated by the nested-grid version of the global 3-D chemical transport model (GEOS-Chem), which included detailed description of oxidant–aerosol chemistry. The model was driven by MERRA-2 assimilated meteorological data (Gelaro et al. 2017). The nested grid over China (15–55°N, 75–135°E) had a horizontal resolution of 0.5° latitude by 0.625° longitude and consisted of 47 vertical layers up to 0.01 hPa. The GEOS-Chem model included the fully coupled O$_3$–NOx–hydrocarbon and aerosol chemistry modules with more than 80 species and 300 reactions (Bey et al. 2001).

Chemical and physical processes were examined using the outputs of GEOS-Chem. Because non-local planetary boundary layer (PBL) mixing was used, emissions and dry deposition trends within the PBL were applied within the mixing (Holtslag and Boville, 1993). Compared with other terms, the value of wet deposition was extremely small, so it was not considered in this study (Liao et al., 2006). Consequently, the major chemical and physical processes related to meteorological conditions included the chemistry, convection, PBL mixing,

transport and their sum within the PBL were the focus.
The GEOS-Chem model has been widely used to examine historical $O_3$ changes in China. Yang et al. (2014)
evaluated the simulated interannual variation of June–July–August (JJA) surface-layer $O_3$ concentration at the
Hok Tsui station (22°13'N, 114°15'E). They found that the model could well capture the peaks and troughs of the
observed JJA $O_3$ concentration with a high correlation coefficient of +0.87 (exceed the 99% confidence level)
between simulations and observations. Moreover, the model could also realistically simulate the spatial
distribution of $O_3$, and the spatial correlation coefficient between simulations and observations in the summer of
2017 could reach up to 0.89 (Li et al. 2019). These studies indicated that the GEOS-Chem model could capture
the interannual variation and distribution of the surface $O_3$ concentration fairly well.
The GEOS-Chem model successfully reproduced the dominant patterns of summer $O_3$ pollution on a daily
scale from 2015 to 2019 (Yin and Ma 2020). In this study, we first simulated the maximum daily average 8 h
concentration of $O_3$ (MDA8 $O_3$) from 2015 to 2019 and evaluated the performance of GEOS-Chem. The simulated
spatial distribution of MDA8 $O_3$ was similar to that of observations with a spatial correlation coefficient of 0.87
(Fig. 1a). Compared the simulated and observed summer mean MDA8 $O_3$ concentrations in NC and the PRD,
which had a low bias with a mean absolute error of 5.7 μg m$^{-3}$ and 12.1 μg m$^{-3}$ in the PRD and NC, respectively.
The values of root mean square error / mean were 15.8 % and 8.1 % in NC and the PRD, respectively. The observed
and simulated summer MDA8 $O_3$ anomalies in the east of China also presented consistent interannual differences
(Fig. S1a, b). The high consistency in both the temporal and spatial distributions between the simulations and
observations provided a solid evidence to support the feasibility of the present study.

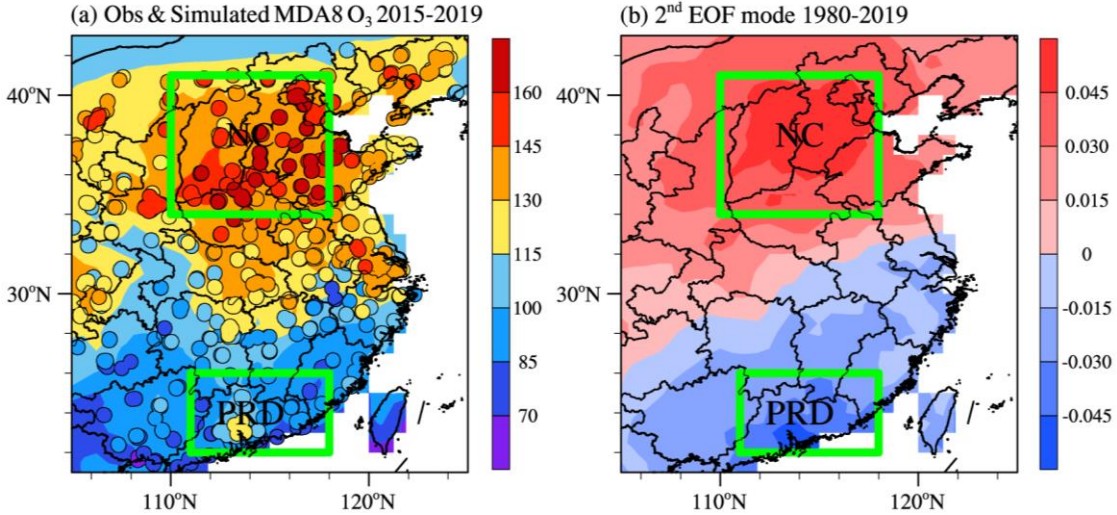

**Figure 1** (a) Spatial distributions of observed (dots) and GEOS-Chem simulated (shading) summer-mean MDA8 $O_3$ (unit: μg
m$^{-3}$) for the period 2015–2019. (b) The second EOF spatial pattern of simulated summer-mean MDA8 $O_3$ from 1980 to 2019.
The simulated $O_3$ concentrations were produced by GEOS-Chem with fixed emissions but changing meteorological conditions
from 1980 to 2019. The green boxes represent the areas of NC and the PRD.
Based the above results, the GEOS-Chem model was then driven by fixed anthropogenic and natural
emissions in 2010 and changing meteorological fields from 1980 to 2019 to highlight the impact of climate
variability on $O_3$ concentration. Results of this simulation were analyzed to reveal the dominant pattern of ozone
pollution in the east of China in summer and its relationship with preceding climate anomalies.
**2.3 Numerical experiments with CESM-LE**
To provide evidences that support the proposed connections between SI and SST and large-scale atmospheric
circulations, the simulations of the Community Earth System Model Large Ensemble (CESM-LE) were employed
(Kay et al. 2015). The CESM consists of coupled atmosphere, ocean, land, and sea ice component models. The
40-member ensemble of CESM-LE simulations over the period (1980–2019) includes a historical simulation
(1980–2005) and a representative concentration pathway (RCP) 8.5 forcing simulation (2006–2019). To confirm
the impact of preceding climate variability and associated physical mechanisms, composite analyses were
conducted based on the three years with the lowest and highest simulated preceding climatic variability for a
particular month in each member. The composite results of atmospheric circulations could be considered as the
relevant atmospheric responses associated with the preceding climate variability.
**3.    Dipole pattern of summer $O_3$ and possible influencing factors**
As aforementioned, the GEOS-Chem model has a good performance in simulating $O_3$ concentration. The
summer $O_3$ concentrations from 1980 to 2019 was simulated by GEOS-Chem, and the EOF approach was applied
to the GEOS-Chem simulation to explore the dominant patterns of summer mean $O_3$ pollution in the east of China.
Percentage contributions to the total variance by the first and second EOF modes were 39% and 17.5%,
respectively. The significance test of the EOF eigenvalues confirmed that the first and second patterns were
distinctly separated (passing the North test, North et al, 1982). The first EOF pattern displayed a monopole pattern
(Fig. S2). The second EOF pattern presented a north-south dipole pattern of $O_3$ (DP-$O_3$) distribution in the east of
China with the two centers located in NC and the Pearl River Delta (PRD, Fig. 1b), respectively. Observations
have shown that high $O_3$ concentration frequently occurs in NC, and $O_3$ pollution in the PRD has become
increasingly serious in recent years (Liu et al. 2020). Furthermore, about 80% of the MDA8 $O_3$ anomalies in NC
were in opposite sign to those in PRD during 2015–2019 (Fig. S1a, b). Therefore, despite the fact that it was only
the second leading EOF mode, we still focused on the investigation of DP-$O_3$ in the present study, since it was
more similar to the actual pollution situation. Impacts of climate variability are also analyzed.

The MDA8 $O_3$ anomalies were divided into positive (P) and negative phases (N) of DP-$O_3$ (Fig. S3). For convenience, DP-$O_3$P and DP-$O_3$N were defined by the EOF time series of DP-$O_3$ greater than 1 standard deviation and less than $-1 \times$ standard deviation, respectively. The DP-$O_3$P corresponded to positive anomalies of MDA8 $O_3$ in the north and negative anomalies in the PRD (Fig. S3a). In contrast, high concentration of $O_3$ occurred in the PRD and low concentration center appeared in NC under the DP-$O_3$N condition (Fig. S3b). The correlation coefficient between time series of DP-$O_3$ and MDA8 $O_3$ difference between NC and the PRD was 0.91, indicating that DP-$O_3$ reflected the opposite changes of $O_3$ concentration in NC and the PRD.

With fixed emissions, the changes in $O_3$ concentrations from 1980 to 2019 were solely caused by meteorological conditions. The time series of DP-$O_3$ showed a strong interannual variation (Fig. 2). Composite differences in large-scale atmospheric circulation and meteorological condition related to DP-$O_3$ between the positive and negative phases (DP-$O_3$P minus DP-$O_3$N) were analyzed to explore the impacts of atmospheric circulation on photochemical reactions and accumulations of various pollutants in the above two areas. During the positive phase of DP-$O_3$, cyclonic and anticyclonic anomalies in the middle troposphere were found over the PRD and NC ($C_{PRD}$ and $AC_{NC}$) (Fig. 3a), respectively. The $C_{PRD}$ and accompanied southerly winds in the PRD efficiently transported clean and moist air from the sea to the PRD (Fig. 3c). Furthermore, low and medium cloud covers were significantly increased, which led to weak solar radiation and reduced photochemical reactions (Fig. 3b). A moist, cool environment and weak solar radiation were conductive to low $O_3$ concentration in the PRD. On the other hand, the positive anomalies of geopotential height in NC increased surface air temperature (Fig. 3a), resulting in a dry environment with decreased cloud covers and sunny weather (Fig. 3b, c).

In order to provide a more quantitative evaluation of the contribution of chemical and physical processes, in Fig. 3d, we examine the area-averaged differences in $O_3$ changes for NC and PRD. Chemistry represents the changes in net chemical production, which appears to be the dominating process, leading to the greatest $O_3$ change between NC and the PRD (12.3 Tons d$^{-1}$, Fig. 3d). Transport represents the change in horizontal and vertical advection of ozone. Depending on the ozone concentration gradient and wind anomalies, the transport difference between NC and PRD is 3.1 Tons d$^{-1}$ (Fig. 3d). Convection changes slightly in NC and PRD. As the mixing process transports ozone along the vertical concentration gradient, it generally contributes negatively to the total ozone change. The above analysis indicates that different meteorological conditions between NC and the PRD led to the difference of $O_3$ concentration in the two regions (differed by 5.2 Tons d$^{-1}$), which eventually contributed the formation of DP-$O_3$.

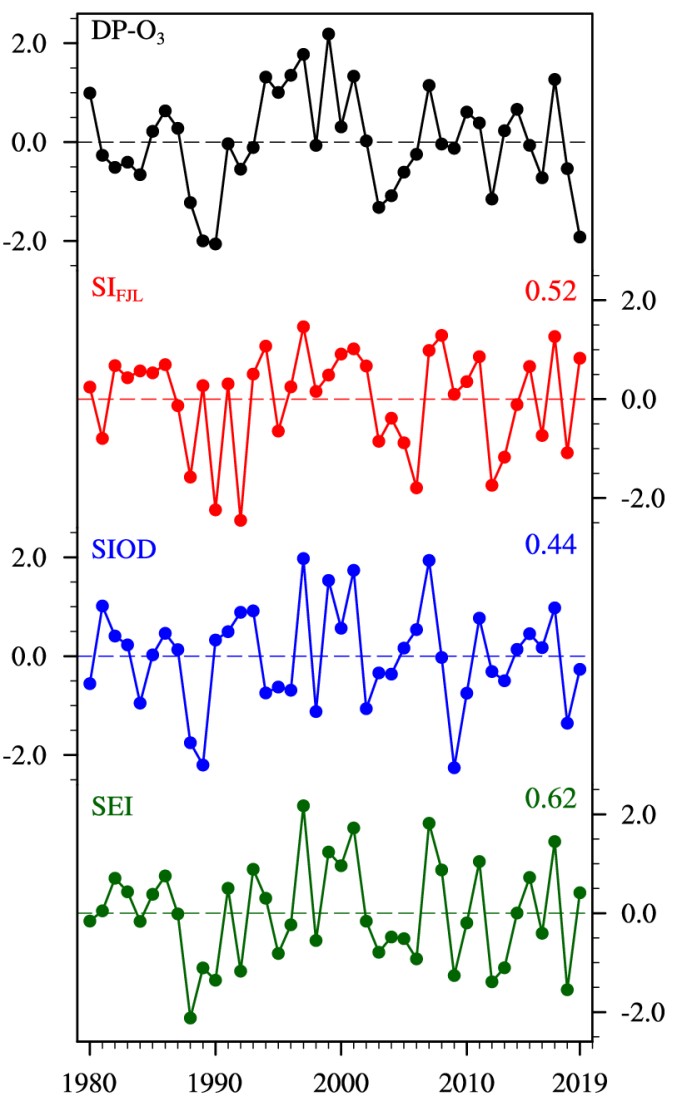

201

**Figure 2** Variations in standardized DP-O$_3$ time series (black), May SI near the Franz Josef Land (SI$_{FJL}$, red), January–
February–March mean Subtropical Indian Ocean Dipole (SIOD, blue), and SEI (green) from 1980 to 2019. SEI defined as the
weighted average of SI$_{FJL}$ and SIOD. The correlation coefficients of the DP-O$_3$ with SI$_{FJL}$ (red), SIOD (blue), and SEI (green)
were shown in the figure.

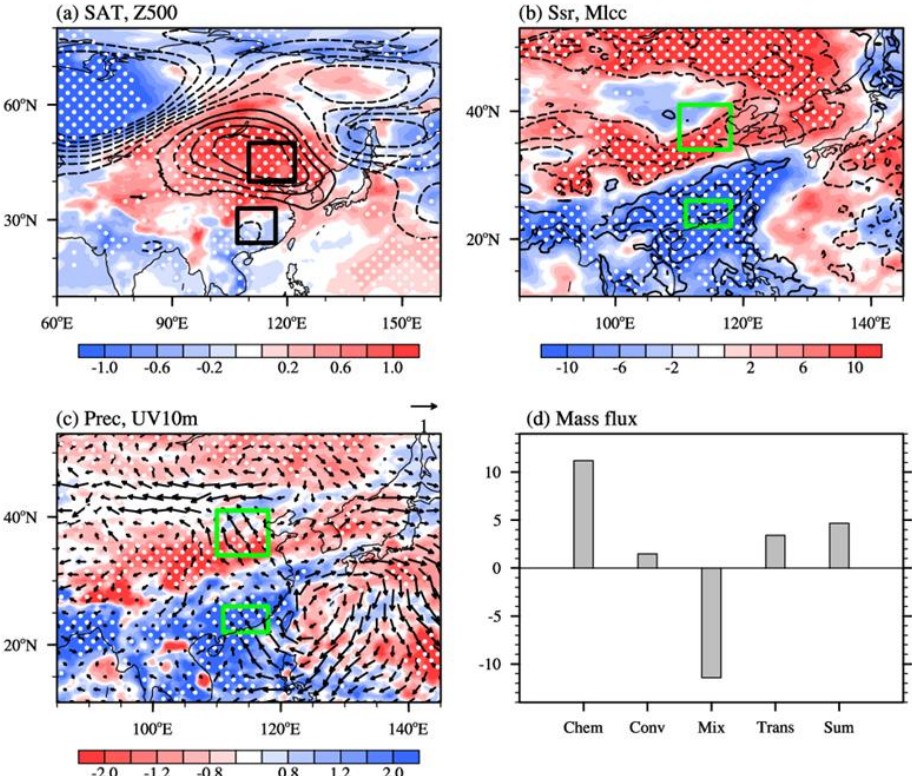

**Figure 3** Composite summer atmospheric circulations associated with the DP-O$_3$ (DP-O$_3$P minus DP-O$_3$N) for the period 1980 to 2019, including (a) surface air temperature (SAT, unit: K, shadings) and geopotential height at 500 hPa (unit: 10 gpm, contours), (b) surface incoming shortwave flux (Ssr, unit: W m$^{-2}$, shadings) and low and medium cloud cover (Mlcc, unit: 1, contours), and (c) precipitation (Prec, unit: mm, shadings) and surface wind (unit: m s$^{-1}$, arrows). The white dots indicate that the composites with shading were above the 90% confidence level. The black boxes in (a) indicate the centers of the AC$_{NC}$ and C$_{PRD}$, respectively. The green boxes in (b) and (c) represent the areas of NC and the PRD. Composites of the summer mass fluxes of O$_3$ (d) associated with the DP-O$_3$ (DP-O$_3$P minus DP-O$_3$N) for the area-averaged differences (NC minus PRD) from 1980 to 2019. The bottom axis gives the names of the chemical and physical processes: chemical reaction (Chem), convection (Conv), PBL mixing (Mix), transport (Trans) and their sum (Sum).

Arctic SI in May was closely related to summer O$_3$ pollution in NC (Yin et al. 2019), but its effects on the north-south dipole distribution of O$_3$ had not been studied. The meridional O$_3$ dipole pattern in the east of China was positively correlated with SI anomalies near the Franz Josef Land (SI$_{FJL}$). Note that the correlation between them remains unchanged after the signal of El Niño-Southern Oscillation (ENSO) was removed. The area-averaged (82–88°N, 3°W–60°E; 79–88°N, 60–90°E; denoted by the green boxes in Fig. 4a) SI in May was calculated and defined as the SI$_{FJL}$ index, whose linear correlation coefficient with the time series of DP-O$_3$ was 0.52 (exceeding the 99% confidence level). When the SI$_{FJL}$ anomalies were significant (i.e., |anomalies| > its one standard deviation), the occurrence probability of the DP-O$_3$ in the same phase was 83% (Fig. 2). Furthermore, the active centers of the anomalous atmospheric circulations and meteorological conditions associated with SI$_{FJL}$ in the east of China were similar to that of the DP-O$_3$ (i.e., NC and PRD). That is, positive SI$_{FJL}$ anomalies were conductive to less (more) precipitation, less (more) cloud cover, and strong (weak) solar radiation in NC (PRD)

(Fig. 4c, Fig. S4). The chemical and physical processes of ozone production in GEOS-Chem simulations were
analyzed. The difference of chemical reactions between NC and PRD had a large positive value (11.6 Tons d$^{-1}$),
and the difference of the sum of all chemical and physical processes was 7.0 Tons d$^{-1}$ (Fig. 4c), resulting in DP-
O$_3$.

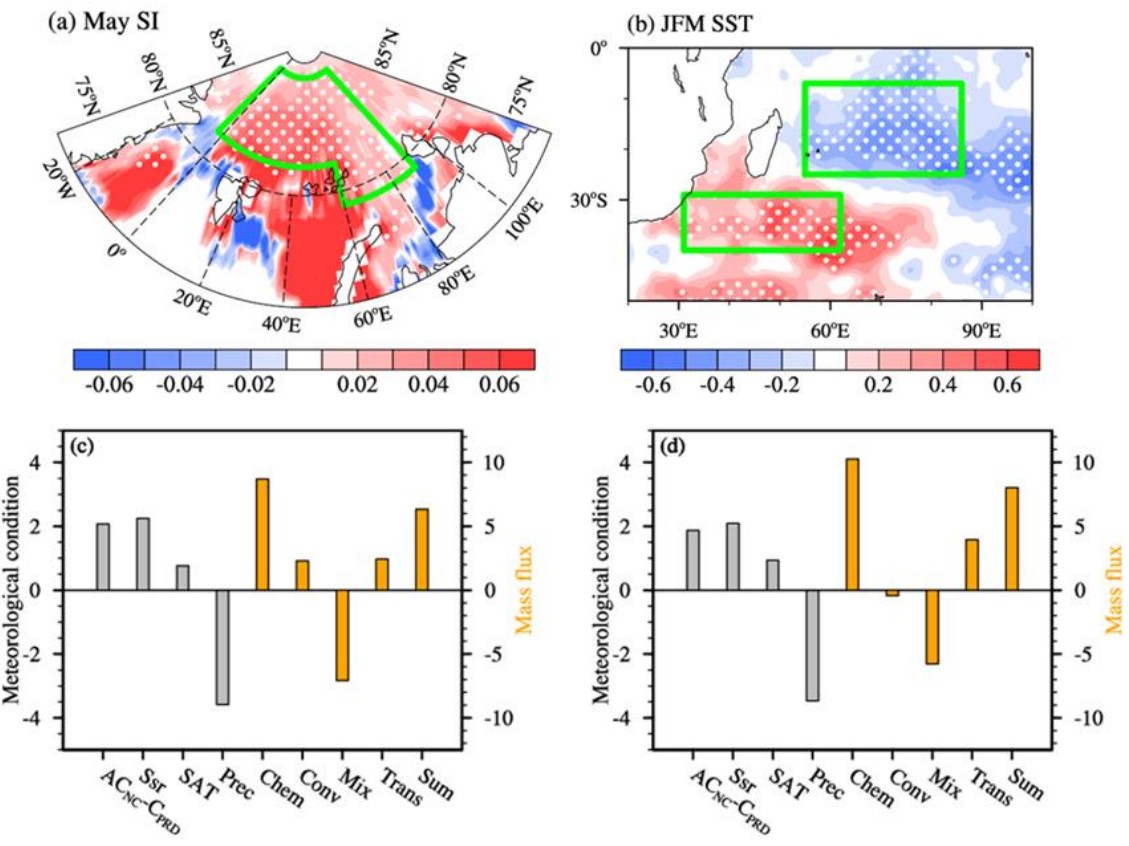


**Figure 4** Composites of (a) May SI concentration and (b) JFM SST associated with the DP-O$_3$ (DP-O$_3$P minus DP-O$_3$N) from
1980 to 2019. The green boxes in (a) and (b) indicate where the SI$_{FJL}$ and SIOD indices are calculated, respectively. The white
dots indicate that the composites were above the 90% confidence level. Composite summer meteorological conditions,
circulations and mass fluxes of O$_3$ associated with (c) SI$_{FJL}$ (positive SI$_{FJL}$ years minus negative SI$_{FJL}$ years) and (d) SIOD
(positive SIOD years minus negative SIOD years) from 1980 to 2019. The bottom axis gives the names of the meteorological
conditions and chemical and physical processes: the differences between AC$_{NC}$ and C$_{PRD}$ (unit: 10 gpm), surface incoming
shortwave flux (Ssr, unit: W m$^{-2}$), surface air temperature (SAT, unit: K), and precipitation (Prec, unit: mm); chemical reaction
(Chem, unit: Tons d$^{-1}$), convection (Conv, unit: Tons d$^{-1}$), PBL mixing (Mix, unit: Tons d$^{-1}$), transport (Trans, unit: Tons d$^{-1}$)
and their sum (Sum, unit: Tons d$^{-1}$).

241       In addition to the signal from the Arctic, SST as an effective external forcing also has significant influences

on summer climate in the east of China (Li et al. 2018). Therefore, it was important to answer the question whether
SST could affect the DP-O$_3$ in the east of China in summer. Large anomalies of preceding January–February–
March (JFM) SST over the southern Indian Ocean was obvious when we evaluated the relationship between the
DP-O$_3$ and previous SST. After removing the influence of ENSO, the SST signal in the southern Indian Ocean
still maintains (Fig. 4b). The two regions with significant anomalies were similar to the Subtropical Indian Ocean
Dipole (SIOD) regions found by Behera and Yamagata (2001). Variance analysis and correlation analysis of SST
in the Indian Ocean also indicated that a SST dipole type oscillation occurred in the southern Indian Ocean, which
usually developed in the preceding winter and reaches its strongest in the subsequent January to March (Jia and
Li 2013). The difference between the mean SST of the two regions (29–40°S, 31–62°E and 7–25°S, 55–86°E;
green box in Fig. 4b; the southwest positive pole minus the northeast negative pole) was defined as the SIOD
index and calculated (Fig. 2). The linear correlation coefficient between the SIOD index and the time series of
DP-$O_3$ from 1980 to 2019 was 0.44 (significant at the 99% confidence level). When the SIOD anomalies were
significant (i.e., |anomalies| > its one standard deviation), the occurrence probability of DP-$O_3$ in the same phase
is 82% (Fig. 2). Furthermore, the composite meteorological conditions in the positive and negative phases of
SIOD had similar centers to that of DP-$O_3$. That is, the anticyclone over NC was always accompanied by hot-dry
meteorological condition, while the cyclone over PRD was always accompanied by cool-moist environment (Fig.
4d; Fig. S5). The chemical reactions increased 12.3 Tons $d^{-1}$ in NC comparing to those in the PRD (Fig. 4 d),
indicating that the strong solar radiation and high temperature conditions actually enhanced the chemical reactions
in the atmosphere to produce more $O_3$ in NC.
**4.  Associated physical mechanisms**
Changes in $SI_{FJL}$ and SIOD both could possibly contribute to the formation of DP-$O_3$. Note that $SI_{FJL}$ and
SIOD have few years of common significant anomalies, more than 78% of the individual sample years were used
to make composite with both indices. The correlation coefficient between them was only 0.21 and was not
significant, indicating that $SI_{FJL}$ and SIOD were independent of each other. Several previous studies have
documented that the preceding Arctic SI anomalies could trigger EU-like atmospheric responses in the subsequent
summer, and thus influenced the climate in the east of China (Wang and He 2015). Corresponding to reduced
$SI_{FJL}$, SST anomalies in the Barents and Kara Sea were significantly positive and gradually increase from May to
summer months (Fig. 5a, b). The warm SST anomalies influenced local heat anomalies and caused anomalous
atmospheric circulations. Following the decrease in $SI_{FJL}$, anomalous divergent winds appeared in the mid-
troposphere, which were accompanied by warm SST anomalies and negative velocity potential anomalies (yellow
box in Fig. 5c). As proposed by Xu et al., (2021), the rotational component of the anomalous divergent winds
could spread to the south and force the vorticity generation over Eurasia. Thus, during the subsequent summer,
significant convergence and positive velocity potential with a positive Rossby wave source anomaly occured over
northern Europe and West Siberia (green box in Fig. 5d). We also used the SST anomalies associated with $SI_{FJL}$
(in Barents and Kara Sea in JJA) to composite relevant variables. Significant convergence, positive velocity
potential, and positive Rossby source anomaly all appeared over Europe and West Siberia in JJA (Fig. S6). This
indicated that positive anomalies of Rossby-wave source over Europe and West Siberia could be generated
by local heat anomalies associated with decreased $SI_{FJL}$ in the Barents and Kara Sea.

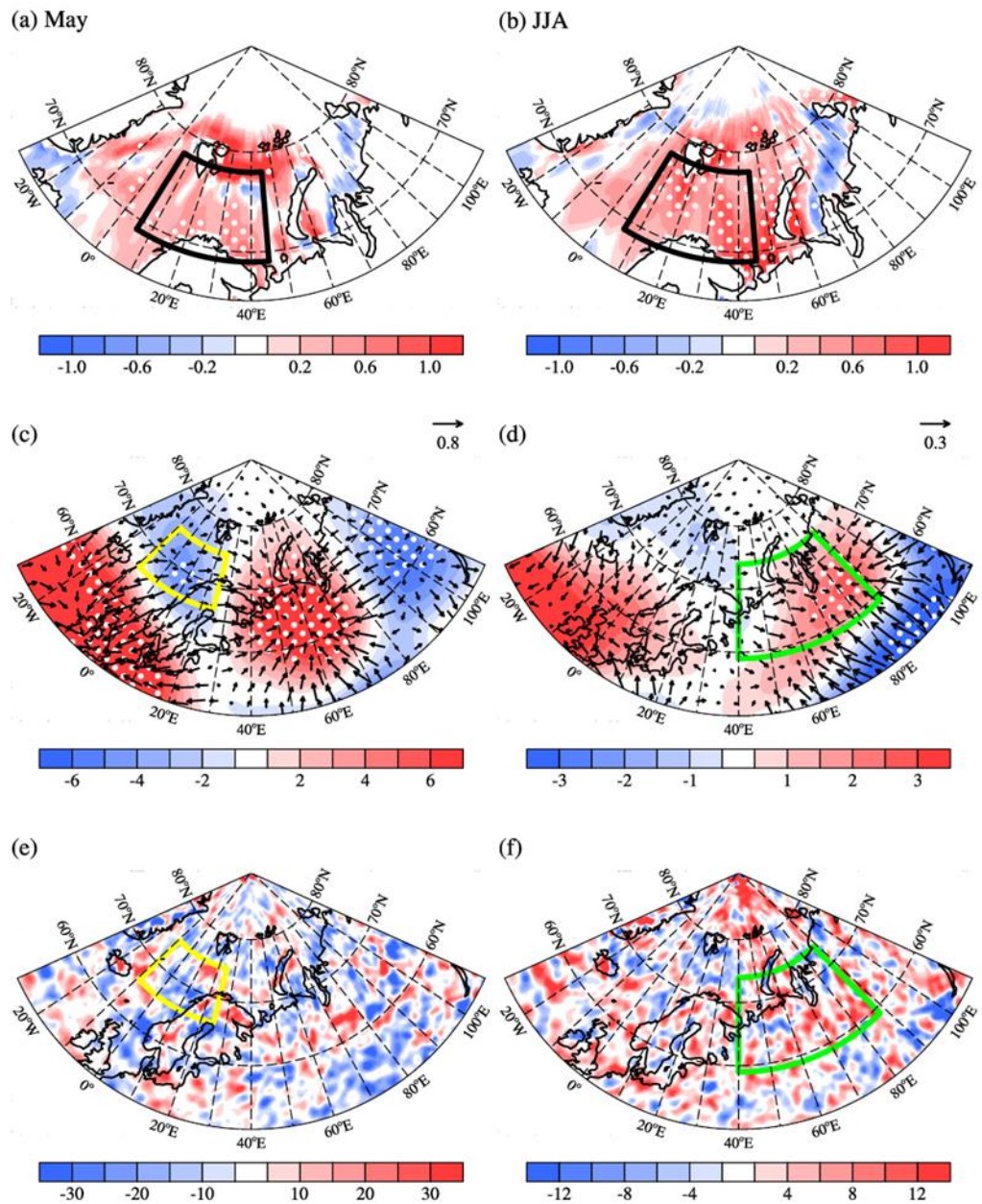


**Figure 5** Composites of (a) May Arctic SST (unit: K), (c) velocity potential (unit: $10^5$ m$^2$ s$^{-1}$, shading) and divergent wind at
500 hPa (unit: m s$^{-1}$, arrows), and (e) Rossby wave source anomalies at 500 hPa (unit: $10^{-11}$ s$^{-2}$) associated with $SI_{FJL}$ index
(negative $SI_{FJL}$ years minus positive $SI_{FJL}$ years) from 1980 to 2019. The back box in (a) and (b), yellow box in (c) and (e) and
green box in (d) and (f) represents the center of the SST, velocity potential and Rossby wave source anomaly associated with
$SI_{FJL}$, respectively. The white dots indicate that the composites with shading were above the 90% confidence level.
Moreover, corresponding to the decreased $SI_{FJL}$, the anomalous Rossby WAF propagated from Europe and
West Siberia (consistent with the aforementioned Rossby wave source) to Northeast China and enhanced the
cyclonic anomaly nearby (Fig. 6a). The anomalous cyclonic circulation caused ascending motion from the surface
up to 300 hPa over NC, and further induced a meridional circulation with an anomalous descending branch near
20°N (Fig. 6b). Likewise, an anomalous anticyclone occurred in the middle troposphere above the PRD (Fig.
6b). In other words, an EU-like Rossby wave train was induced in the mid-troposphere (Fig. 6a), which
propagated from northern Europe and West Siberian Plain (+), reaching the broad area from northeastern China
(–) to the south of China (+). Thus, the reduction in SI near the Franz Josef Land in the May modulated the EU-
like pattern in the subsequent summer and strengthened the anomalous cyclonic and anticyclonic circulations over
NC and the PRD (Fig. 6b), respectively. The differences in anomalous atmospheric circulations and associated
meteorological conditions between NC and the PRD make great contributions to the occurrence of DP-O$_3$.

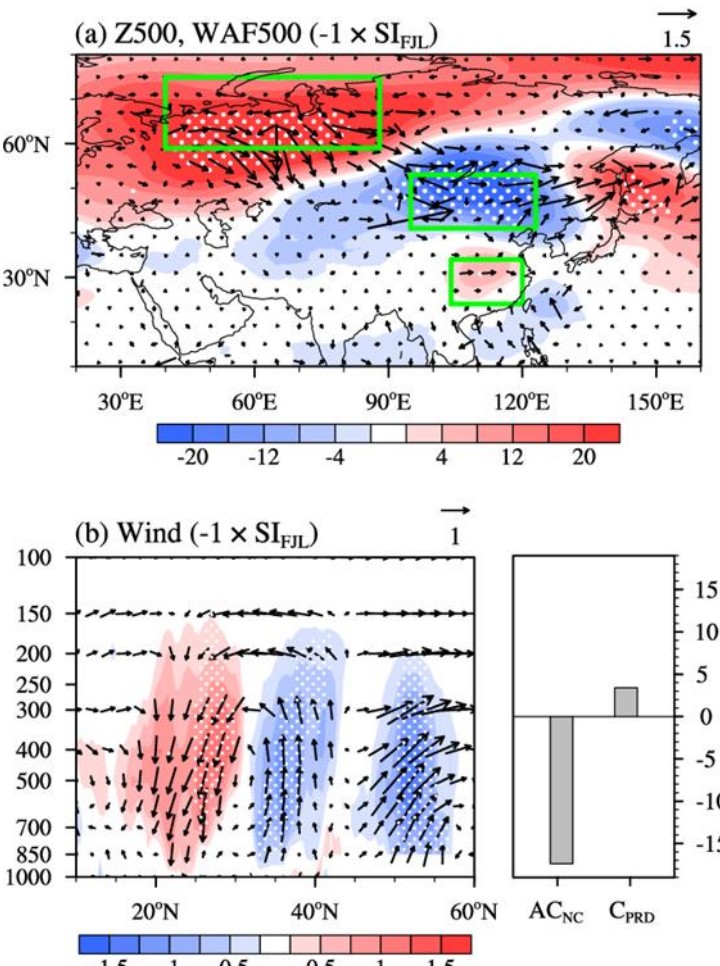


**Figure 6** Composites of (a) wave activity flux anomalies (unit: m$^2$ s$^{-2}$, arrows), geopotential height (unit: gpm, shading) at 500
hPa and (b) mean wind (unit: m s$^{-1}$, arrows), omega (unit: 10$^{-2}$ Pa s$^{-1}$, shading) over 100–130° E, and the anomalies of AC$_{NC}$
and C$_{PRD}$ (unit: gpm, bar) in summer associated with SI$_{FJL}$ index (negative SI$_{FJL}$ years minus positive SI$_{FJL}$ years) from 1980
to 2019. The green boxes in (a) represent the centers of the EU-like pattern. The white dots indicate that the composites with
shading were above the 90% confidence level.
The relationship between the preceding May SI anomalies and the JJA EU-like pattern was also confirmed
by large ensemble simulations of CESM during 1980–2019. According to the simulated sea ice fraction near the
Franz Josef Land, the three years with the lowest and highest SI in each member were selected to construct the

composite maps based on all the 40 available members. The difference in JJA geopotential height at 500 hPa represented the atmospheric response to declining May $SI_{FJL}$. As shown in Fig. 7, the decline of $SI_{FJL}$ in May led to an EU-like pattern in the subsequent summer over Eurasia, which was in good accordance with the observed result (Fig. 6a). The anticyclonic and cyclonic anomalies shown in the geopotential height at 500 hPa (i.e., $AC_{NC}$ and $C_{PRD}$) in summer were also well reproduced by over 60% of the members. The above results confirmed the robustness of the physical mechanisms proposed in the present study.

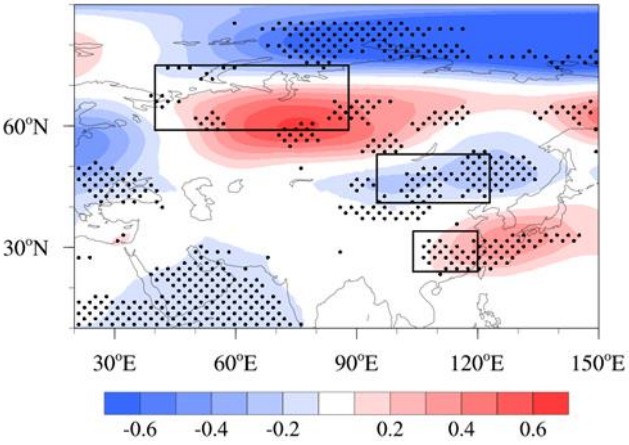

**Figure 7** Composite differences of geopotential height at 500 hPa in JJA between three low and high $SI_{FJL}$ years based on the ensemble of 40 CESM-LE simulations during 1980–2019. The black dots indicate that the mathematical sign of the composite results of more than 60 % of the members is consistent with the ensemble mean. The black boxes represent the centers of the EU-like pattern.

SIOD could influence atmospheric anomalies and distribution of summer precipitation in China mainly through Hadley circulation (Liu et al. 2019). Can SIOD anomalies also influenced the $DP$-$O_3$ via meridional atmospheric forcing? Despite the significant correlation between SIOD anomalies (defined by SST) and the $DP$-$O_3$ in the east of China (Fig. 4b), it should be noted that the thermodynamic signals in the southern Indian Ocean not only existed on the sea surface but also extended to the subsurface (Fig. S7). As time goes by, the center of negative SST anomalies moved to the northeast possibly due to the eastward movement of atmospheric forcing caused by the mean westerly flow (Behera and Yamagata 2001). When it moved to the vicinity of Sumatra Island in JJA, the abnormally cold signals of SST could extend downward from the surface to 60m (black box in Fig. 8a). The area-averaged (black box in Fig. 8a) summer-mean subsurface ocean temperature of 0–60m was defined as the SOT index and calculated. Affected by negative SOT anomalies near Sumatra Island, the equatorial eastern Indian Ocean convection was suppressed (indicated by positive anomalies of OLR in Fig. 8b) and significant divergence prevailed in the lower troposphere (Fig. 8c). As a result, anomalous downward air flow developed near Sumatra Island from 300 hPa to the surface (about 20–5°S in Fig. 8d). This anomalous downward air flow modulated the meridional circulation over 90–120 °E by strengthening the abnormal upward airflow at 20°N and

downward airflow at 30°N. Thus, the $AC_{NC}$ and $C_{PRD}$ were enhanced simultaneously (Fig. 8d). Overall, following

the positive phase of SIOD, the cold signal of SOT anomalies changed the meridional circulation in the subsequent

JJA and strengthened the $C_{PRD}$ and $AC_{NC}$ in the troposphere above the east of China. Under these large-scale

atmospheric anomalies, $O_3$ concentrations became higher in NC, whereas the generation of surface $O_3$ were

weakened in the PRD.

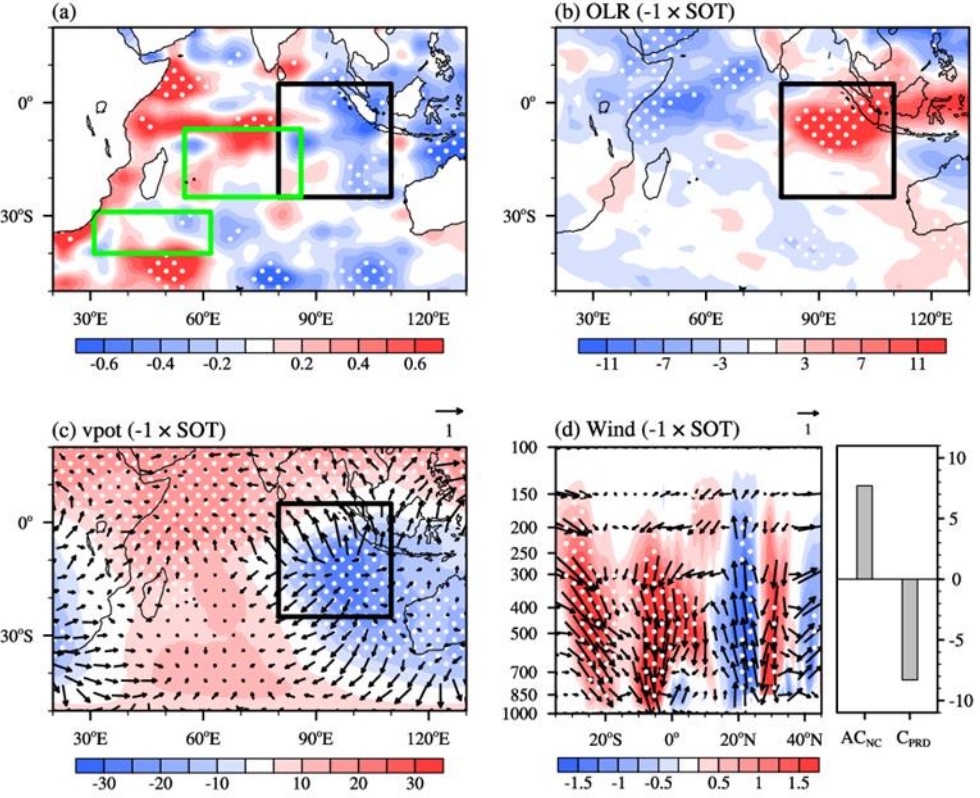

**Figure 8** (a) Composites of mean 0–60m subsurface ocean temperature (unit: K) in summer associated with the SIOD (positive

SIOD years minus negative SIOD years) from 1980 to 2019. The green boxes represent the centers of the SIOD, and the black

box indicates where the SOT index is calculated. Composites of (b) OLR (unit: W m$^{-2}$) and (c) velocity potential (unit: $10^5$ m$^2$

s$^{-1}$, shadings) and divergent winds (unit: m s$^{-1}$, vectors) at 10 m in summer associated with SOT indexes of opposite sign

(negative SOT years minus positive SOT years). The black box represents the center of the SOT. (d) Composites of summer

mean winds (unit: m s$^{-1}$, arrows) and omega (unit: $10^{-2}$ Pa s$^{-1}$, shadings) over 90–120°E, and the anomalies of $AC_{NC}$ and $C_{PRD}$

(unit: gpm, bars) associated with SOT indexes of opposite sign. The white dots indicate that the composites with shading were

above the 90% confidence level.

The CESM-LE datasets were also used to verify the statistical correlation between the preceding SIOD and

large-scale atmospheric circulations in JJA. The composite differences of SIOD in JFM between the three high

years and three low years of SST simulated by each ensemble member during 1980–2019 were investigated based

on the ensemble of 40 CESM-LE simulations. The composite results (positive SIOD years minus negative SIOD

years) of atmospheric circulations could be considered as the relevant atmospheric circulation responses

associated with differences in SIOD. More than 60% of the CESM ensemble members could well reproduce the

anticyclonic circulation over NC and the cyclonic circulation over the PRD in summer at 500hPa (Fig. 9). That is,
the CESM-LE also confirmed the relationship between the previous JFM SIOD anomaly and the DP-$O_3$-related
atmospheric circulations (i.e., $AC_{NC}$ and $C_{PRD}$) in subsequent JJA.

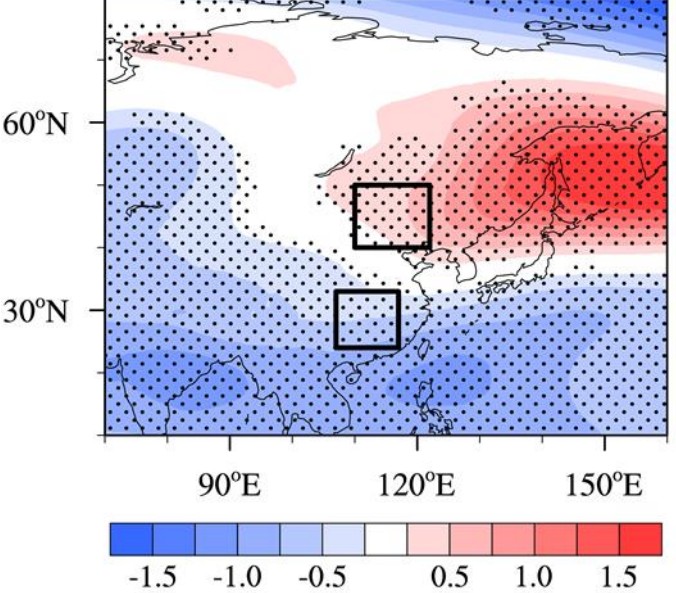


**Figure 9** Composite differences of geopotential height at 500 hPa in JJA between three high and low SIOD years based on the
ensemble of 40 CESM-LE simulations during 1980–2019. The black dots indicate that the mathematical sign of the composite
results of more than 60 % of the members is consistent with the ensemble mean. The black boxes represent the centers of $AC_{NC}$
and $C_{PRD}$, respectively.

## 5.    Conclusions and discussions

In general, the $O_3$ concentrations in NC were substantially high and the problem of $O_3$ pollution in the PRD
has become increasingly prominent in recent years. A south-north dipole pattern of $O_3$ concentration in the east of
China was identified based on GEOS-Chem simulations with fixed emissions and changing meteorological
condition from 1980 to 2019. The DP-$O_3$ pattern presented opposite centers in NC and PRD. Corresponding to
the positive phase of DP-$O_3$, cyclonic and anticyclonic anomalies were located over the PRD and NC respectively,
which resulted in dry and hot climate in NC, while the environment in the PRD region was cool and moist. The
opposite was true in the negative phase of DP-$O_3$. During positive phases, the meteorological condition mentioned
above significantly enhanced photochemical reactions in NC but suppressed $O_3$ production in the PRD, and thus
make great contributions to the south-north dipole pattern of $O_3$ in the east of China.
Arctic SI near the Franz Josef Land in May played an important role in the occurrence of DP-$O_3$. The warm
SST anomalies associated with less $SI_{FJL}$ could induce divergent wind field and vorticity advection in the upper
layer, and enhanced positive Rossby wave source over northern Europe and West Siberia in summer. An EU-like
pattern was triggered in Eurasia (solid lines in Fig. 10), which could enhance the DP-O$_3$-related atmospheric
circulation (i.e., AC$_{NC}$ and C$_{PRD}$) in JJA. As a result, meteorological conditions for O$_3$ concentration were
completely different between NC and PRD, which eventually contributed the formation of DP-O$_3$. In addition, the
precursory climatic driving signal of SIOD anomalies in the low latitudes in JFM was also closely linked to DP-
O$_3$. The thermodynamic signal of SIOD could be stored in the subsurface, and the center of negative SST
anomalies moved to the vicinity of Sumatra Island in summer. The meridional circulation intensified in summer
(dashed lines in Fig. 10), which, along with the enhancement of the AC$_{NC}$ and C$_{PRD}$ over the east of China,
effectively increased O$_3$ concentration in NC but suppressed the generation of surface O$_3$ in the PRD. The linkages
and corresponding physical mechanisms were well reproduced by the large CESM-LE ensemble simulation.

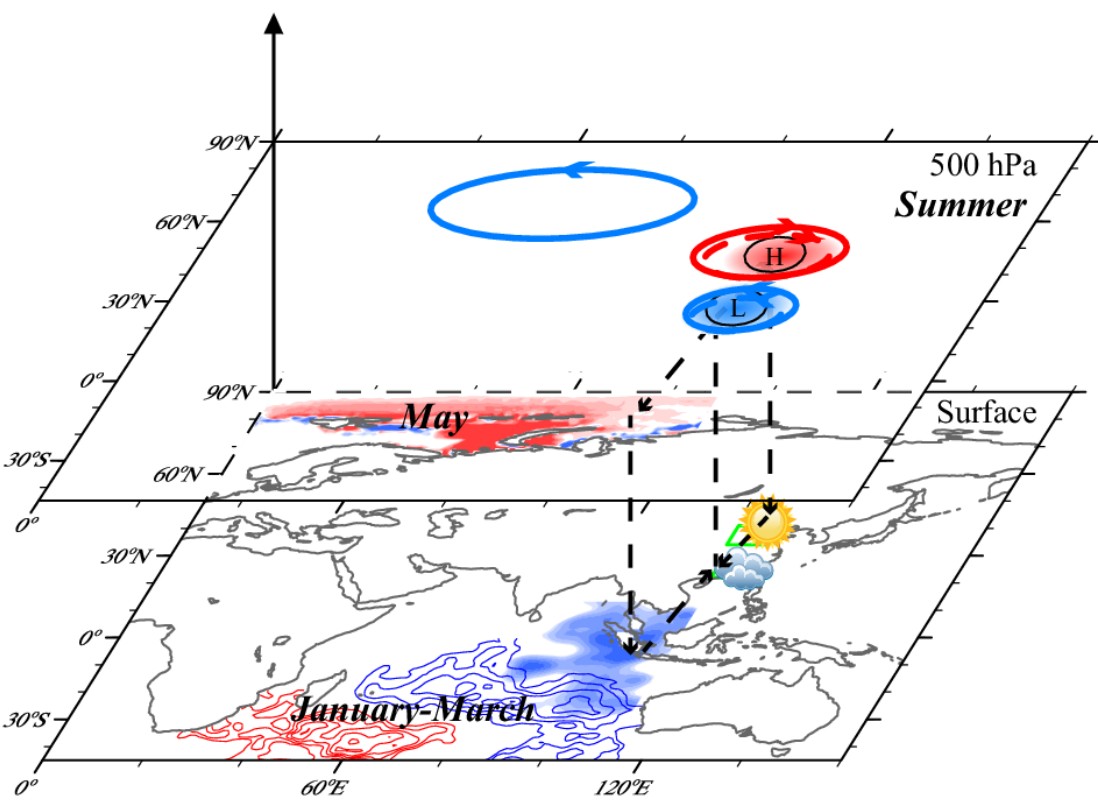


**Figure 10** Schematic diagrams of the associated physical mechanisms. The May SI anomalies near the Franz Josef Land (red shadings) could trigger an EU-like pattern in the atmosphere in summer, which enhances the anticyclonic anomaly over NC and the cyclonic anomaly over the PRD. The thermodynamic signal of the preceding SIOD (contours) could be stored in the subsurface and the center of negative SST anomalies moves to the vicinity of Sumatra Island in summer (blue shading). The meridional circulation was enhanced in summer (dashed lines), along with the enhancement of AC$_{NC}$ and C$_{PRD}$ over eastern China. The solid lines indicate the anomalous atmospheric circulations affected by SI$_{FJL}$, while the dashed lines indicate the anomalous atmospheric circulations affected by SIOD.

The above analysis has revealed that the DP-O$_3$ is independently affected by SIOD and SI$_{FJL}$ from 1980 to
2019. We attempted to discuss the combined impacts of the two precursory climatic drivers in the present
study. For this purpose, a synthetic climate variability index SEI, defined as the weighted average of SI$_{FJL}$ and
SIOD, is calculated by
$$SEI = \frac{r_1 \times SI_{FJL} + r_2 \times SIOD}{|r_1| + |r_2|}$$

where $r_1$ and $r_2$ were the correlation coefficients of $SI_{FJL}$ ($r_1 = 0.52$) and SIOD ($r_2 = 0.44$) with the DP-O$_3$
time series, respectively. The correlation coefficient between SEI and DP-O$_3$ was 0.62 (Fig. 2, exceeding the
99% confidence level). When the SEI anomalies were significant, the occurrence probability of the DP-O$_3$ in the
same phase was 93% (Fig. 2), which is higher than that based on individual influences of the two factors.
Composite atmospheric circulation analysis has been carried out based on years of positive and negative SEI
anomalies, and the results are shown in Fig. 11a. The composite atmospheric circulation based on the SEI index
was stronger, resulting in the concentrations of MDA8 O$_3$ in NC was 11.74 μg m$^{-3}$ higher than that in PRD (Fig.
11b). The main areas influenced by SI and SST were slightly different. Although the two precursory climatic
drivers both could affect the atmospheric circulations over NC and the PRD, $SI_{FJL}$ mainly affected atmospheric
circulation anomaly over NC, while SIOD played a major role in the PRD. However, climate variabilities at
different latitudes jointly facilitated the dipole pattern of O$_3$ in the east of China from 1980 to 2019.

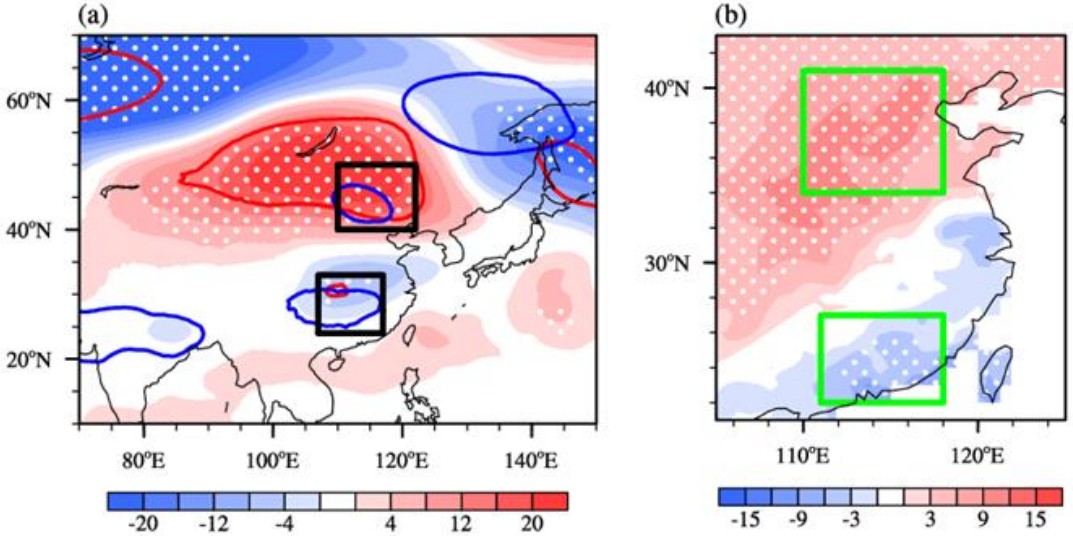


**Figure 11** (a) Composites of geopotential height at 500 hPa (unit: gpm, shadings) in summer associated with the SEI (positive
SEI years minus negative SEI years) from 1980 to 2019. The red and blue lines indicate areas where the composite
geopotential height anomalies associated with $SI_{FJL}$ and SIOD exceed the 90% confidence level, respectively. The black boxes
represent the centers of AC$_{NC}$ and C$_{PRD}$, respectively. (b) Composite differences of the detrended summer-mean MDA8 O$_3$
(unit: μg m$^{-3}$) simulated by GEOS-Chem model between high and low SEI years during 1980–2019. The white dots indicate
that the composite differences are above the 90% confidence level. The green boxes represent the areas of NC and the PRD.
The north-south dipole pattern of O$_3$ in the east of China in summer and its relationship with climate factors
were clearly revealed in this study, yet some questions still remain unanswered and should be investigated in the
future. The GEOS-Chem model simulations were used to explore the dominant pattern of O$_3$ in the east of China
in summer due to the short sequence of O$_3$ observations. Although the GEOS-Chem demonstrated a good
performance based on evaluation, there still exist some differences between the simulations and observations. In
addition, statistical and numerical methods were used to reveal and verify the physical mechanisms behind the
dipole pattern of O$_3$ in the east of China and its relation with climate variability. However, further numerical
experiments should be carried out in the future. For example, coupled climate-chemistry models should be used
to not only simulated the influence of climate driving factors on O$_3$ pattern, but also revealed the effect of
individual climate factors as well as their comprehensive effects.

**Data Availability.**
Hourly O$_3$ concentration data could be downloaded from https://quotsoft.net/air/ (Ministry of Environmental
Protection of China, 2020). Sea ice concentration, sea surface temperature, and subsurface ocean temperature data
were from https://www.metoffice.gov.uk/hadobs/ (Met Office Hadley Centre, 2021). Monthly-mean MERRA-2
reanalysis dataset was available at https://disc.gsfc.nasa.gov/datasets?page=1 (MERRA-2, 2021). The monthly
OLR data could be acquired from http://olr.umd.edu/ (University of Maryland OLR Climate Data Record portal,

429 2021).


**Authors' contribution**
Yin Z. C. designed the research. Ma X. Q. did the statistical analysis and implemented the GEOS-Chem
simulations. Yin Z. C. and Ma X. Q. prepared the manuscript.

**Competing interests**
The authors declare that they have no conflict of interest.

**Acknowledgements**
This research was supported by National Natural Science Foundation of China (42025502, 42088101, 41991283
and 91744311).

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

**Table and Figure captions**

**Figure 1** (a) Spatial distributions of observed (dots) and GEOS-Chem simulated (shading) summer-mean MDA8 $O_3$ (unit: $\mu g\ m^{-3}$) for the period 2015–2019. (b) The second EOF spatial pattern of simulated summer-mean MDA8 $O_3$ from 1980 to 2019. The simulated $O_3$ concentrations were produced by GEOS-Chem with fixed emissions but changing meteorological conditions from 1980 to 2019. The green boxes represent the areas of NC and the PRD.

**Figure 2** Variations in standardized DP-$O_3$ time series (black), May SI near the Franz Josef Land ($SI_{FJL}$, red), January–February–March mean Subtropical Indian Ocean Dipole (SIOD, blue), and SEI (green) from 1980 to 2019. SEI defined as the weighted average of $SI_{FJL}$ and SIOD. The correlation coefficients of the DP-$O_3$ with $SI_{FJL}$ (red), SIOD (blue), and SEI (green) were shown in the figure.

**Figure 3** Composite summer atmospheric circulations associated with the DP-$O_3$ (DP-$O_3$P minus DP-$O_3$N) for the period 1980 to 2019, including (a) surface air temperature (SAT, unit: K, shadings) and geopotential height at 500 hPa (unit: 10 gpm, contours), (b) surface incoming shortwave flux (Ssr, unit: $W\ m^{-2}$, shadings) and low and medium cloud cover (Mlcc, unit: 1, contours), and (c) precipitation (Prec, unit: mm, shadings) and surface wind (unit: $m\ s^{-1}$, arrows). The white dots indicate that the composites with shading were above the 90% confidence level. The black boxes in (a) indicate the centers of the $AC_{NC}$ and $C_{PRD}$, respectively. The green boxes in (b) and (c) represent the areas of NC and the PRD. Composites of the summer mass fluxes of $O_3$ (d) associated with the DP-$O_3$ (DP-$O_3$P minus DP-$O_3$N) for the area-averaged differences (NC minus PRD) from 1980 to 2019. The bottom axis gives the names of the chemical and physical processes: chemical reaction (Chem), convection (Conv), PBL mixing (Mix), transport (Trans) and their sum (Sum).

**Figure 4** Composites of (a) May SI concentration and (b) JFM SST associated with the DP-$O_3$ (DP-$O_3$P minus DP-$O_3$N) from 1980 to 2019. The green boxes in (a) and (b) indicate where the $SI_{FJL}$ and SIOD indices are calculated, respectively. The white dots indicate that the composites were above the 90% confidence level. Composite summer meteorological conditions, circulations and mass fluxes of $O_3$ associated with (c) $SI_{FJL}$ (positive $SI_{FJL}$ years minus negative $SI_{FJL}$ years) and (d) SIOD (positive SIOD years minus negative SIOD years) from 1980 to 2019. The bottom axis gives the names of the meteorological conditions and chemical and physical processes: the differences between $AC_{NC}$ and $C_{PRD}$ (unit: 10 gpm), surface incoming shortwave flux (Ssr, unit: W $m^{-2}$), surface air temperature (SAT, unit: K), and precipitation (Prec, unit: mm); chemical reaction (Chem, unit: Tons $d^{-1}$), convection (Conv, unit: Tons $d^{-1}$), PBL mixing (Mix, unit: Tons $d^{-1}$), transport (Trans, unit: Tons $d^{-1}$) and their sum (Sum, unit: Tons $d^{-1}$).

**Figure 5** Composites of (a) May Arctic SST (unit: K), (c) velocity potential (unit: $10^5$ $m^2$ $s^{-1}$, shading) and
divergent wind at 500 hPa (unit: m $s^{-1}$, arrows), and (e) Rossby wave source anomalies at 500 hPa (unit: $10^{-11}$ $s^{-2}$)
associated with $SI_{FJL}$ index (negative $SI_{FJL}$ years minus positive $SI_{FJL}$ years) from 1980 to 2019. The back box in
(a) and (b), yellow box in (c) and (e) and green box in (d) and (f) represents the center of the SST, velocity potential
and Rossby wave source anomaly associated with $SI_{FJL}$, respectively. The white dots indicate that the composites
with shading were above the 90% confidence level.
**Figure 6** Composites of (a) wave activity flux anomalies (unit: $m^2$ $s^{-2}$, arrows), geopotential height (unit: gpm,
shading) at 500 hPa and (b) mean wind (unit: m $s^{-1}$, arrows), omega (unit: $10^{-2}$ Pa $s^{-1}$, shading) over 100–130° E,
and the anomalies of $AC_{NC}$ and $C_{PRD}$ (unit: gpm, bar) in summer associated with $SI_{FJL}$ index (negative $SI_{FJL}$ years
minus positive $SI_{FJL}$ years) from 1980 to 2019. The green boxes in (a) represent the centers of the EU-like pattern.
The white dots indicate that the composites with shading were above the 90% confidence level.
**Figure 7** Composite differences of geopotential height at 500 hPa in JJA between three low and high $SI_{FJL}$ years
based on the ensemble of 40 CESM-LE simulations during 1980–2019. The black dots indicate that the
mathematical sign of the composite results of more than 60 % of the members is consistent with the ensemble
mean. The black boxes represent the centers of the EU-like pattern.
**Figure 8** (a) Composites of mean 0–60m subsurface ocean temperature (unit: K) in summer associated with the
SIOD (positive SIOD years minus negative SIOD years) from 1980 to 2019. The green boxes represent the centers
of the SIOD, and the black box indicates where the SOT index is calculated. Composites of (b) OLR (unit: W $m^{-2}$)
and (c) velocity potential (unit: $10^5$ $m^2$ $s^{-1}$, shadings) and divergent winds (unit: m $s^{-1}$, vectors) at 10 m in summer
associated with SOT indexes of opposite sign (negative SOT years minus positive SOT years). The black box
represents the center of the SOT. (d) Composites of summer mean winds (unit: m $s^{-1}$, arrows) and omega (unit:
$10^{-2}$ Pa $s^{-1}$, shadings) over 90–120°E, and the anomalies of $AC_{NC}$ and $C_{PRD}$ (unit: gpm, bars) associated with SOT
indexes of opposite sign. The white dots indicate that the composites with shading were above the 90% confidence
level.
**Figure 9** Composite differences of geopotential height at 500 hPa in JJA between three high and low SIOD years
based on the ensemble of 40 CESM-LE simulations during 1980–2019. The black dots indicate that
mathematical sign of the composite results of more than 60 % of the members is consistent with the ensemble
mean. The black boxes represent the centers of $AC_{NC}$ and $C_{PRD}$, respectively.
**Figure 10** Schematic diagrams of the associated physical mechanisms. The May SI anomalies near the Franz Josef
Land (red shadings) could trigger an EU-like pattern in the atmosphere in summer, which enhances the
anticyclonic anomaly over NC and the cyclonic anomaly over the PRD. The thermodynamic signal of the
preceding SIOD (contours) could be stored in the subsurface and the center of negative SST anomalies moves to
the vicinity of Sumatra Island in summer (blue shading). The meridional circulation was enhanced in summer
(dashed lines), along with the enhancement of $AC_{NC}$ and $C_{PRD}$ over eastern China. The solid lines indicate the
anomalous atmospheric circulations affected by $SI_{FJL}$, while the dashed lines indicate the anomalous atmospheric
circulations affected by SIOD.
**Figure 11** (a) Composites of geopotential height at 500 hPa (unit: gpm, shadings) in summer associated with the
SEI (positive SEI years minus negative SEI years) from 1980 to 2019. The red and blue lines indicate areas where
the composite geopotential height anomalies associated with $SI_{FJL}$ and SIOD exceed the 90% confidence level,
respectively. The black boxes represent the centers of $AC_{NC}$ and $C_{PRD}$, respectively. (b) Composite differences of
the detrended summer-mean MDA8 $O_3$ (unit: $\mu g\ m^{-3}$) simulated by GEOS-Chem model between high and low
SEI years during 1980–2019. The white dots indicate that the composite differences are above the 90% confidence
level. The green boxes represent the areas of NC and the PRD.