# Peer review of "Figure S1. Histogram of JJA mean observed (gray) and simulated (orange) MDA8 O3 after detrending in NC (a) and PRD (b) from 2015 to 2019. (c) Spatial distribution of GEOS-Chem simulated MDA8 O3 (unit: $\mu\text{g m}^{-3}$ ) in summer from 1980 to 2019. The simulated O"

_Atmospheric Chemistry and Physics, 2021_

## Referee Comment (RC1)

Review report: ACPD Ma and Yin (2021)

This manuscript discusses the dipole like pattern in summer time ozone concentrations (DPO) seen in the north and south parts of China. The authors particularly focus on the roles of interannual variability (IAV) of climate dynamics like Rossby wave activity and propagation, and Indian ocean SST (dipole mode). Since $O_3$ pollution, being increasingly important, is now one of the biggest issues of air quality as well as PM2.5 which is on the decrease in China, revealing underlying mechanisms of interannual changes and variability in surface O3 including precursor emissions and meteorological factors is of importance. I enjoyed much their discussions on the mechanisms that cause dipole anomaly patterns over China focusing on the roles of Arctic sea-ice and southern Indian ocean SSTs. However, the authors do seem to only focus on the aspect of climate dynamics, just looking at the meteorological parameters like pressure field, precipitation, and fail to address the actual mechanism of the dipole like O3 anomalies over China. They do not seem to answer enoughly to the important questions of how meteorological changes give a rise surface O3 anomalies in the region (upward/horizontal transport? Chemical loss? Chemical production?). Furthermore, I do not fully agree with the analyzing approach adopted in this manuscript. For the O3 variation, they use simulation with GEOS-CHEM, while for meteorological/climate parameters, another dataset like Era-5 is used instead. To keep consistency, the authors should have used the same meteorology as used in GEOS-CHEM. I rate this manuscript as "major revision" at this moment, but think this stuff could be published in ACP after additional analysis and modification are properly included.

**Major comments:**

**M1)** The authors use different meteorological datasets for ozone (DPO) and meteorological parameters (i.e. GEOS-Chem with MERRA-2, and ERA-5), isn't it possible to fix this inconsistency by synchronizing those two? Ideally, you should close the overall discussions within the framework of GEOS-Chem model simulation. At least, they should verify consistency between the Era-5 and MERRA2 in detail.

**M2)** The DPO patterns are only evaluated with surface observations in China (Figure 1). But the DPO pattern could and should be further evaluated by using satellite observation like OMI/TROPOMI. This could make your later discussions (relationship with SI or SIOD) further clear and robust.

**M3)** As pointed out above, you should more clearly discuss the actual mechanism of DPO. At least, is should be separated into two categories: 1) transport (bot vertical and horizontal motion could affect surface O3), and 2) chemistry (production and net-production of O3 in the regions). You should evaluate the chemical roles by taking a look at chemical tendency (P and P-L) of O3 in GEOS-Chem. For transport aspect, this can be evaluated by focusing on the concurrent changes in distribution of carbon monoxide (CO) or any other inert-like tracer. And if you look at CO, satellite observation (MOPITT etc.) may further

help and reinforce the discussion on the transport aspects.

**M4)** You need describe more about your GEOS-Chem simulations especially for the emissions. How does the model consider natural emissions like BVOCs and LNOx? In this manuscript, the authors state that they use GEOS-Chem simulations with fixed emissions. What kind of emission types are targeted for this treatment? (only anthropogenic or natural emissions also?) If the natural emissions are not fixed and follow the climate/meteorological variability, the authors should add description on how these natural emission IAVs affect DPO. On the other hand, if the natural emissions are fixed as anthropogenic ones, you should in turn discuss how the natural emissions, if they are allowed to vary in response to climate condition, would cause additional effects.

**M5)** For the dipole pattern over N/S China, the authors only try to attribute it to SI and SIOD. But in fact, this phenomenon should be also tightly linked to Asian monsoon and ENSO, shouldn't it? Please extend your discussions to cover the monsoon and ENSO influences.

**Minor comments:**

1) L41: "were lower" → "were lower by about *** ppbv"
2) L46: "enhancement of natural emissions of ozone precursors" Yes it's true, but it is not clear at all how natural emissions are treated in this study.
3) L98-103: The math formulation should be further explained. What are $f$, $\zeta$, $U$? and what do "_x", "_xy", and prime "'" indicate? (I know what they are, but major part of readers can not catch them on)
4) L136: Doesn't this version of CESM-LE consider chemistry? If so, you could discuss DPO in CESM—LE as well.
5) L147-148: "the GEOS-Chem model has a good performance … Therefore EOF was applied" : I don't understand the logic here. The sentences should be rephrased.
6) L152: "the first EOF pattern …": What does this 1st EOF mode stand for as meteorological phenomenon?
7) L176: "a moist, cool, … weak solar radiation were conductive to low O3" : Please check and discuss changes in chemical production and photochemical lifetime of O3 in GEOS-Chem.
8) L192: ", but its effected" → ", but its effects"
9) L216: "After removing the influences of ENSO": What are the ENSO influences like? And how did you remove them?
10) L226: "82%" how did you draw this value?
11) L227: "active centers" I didn't follow this.
12) L232: Note that the correlation coefficient between them was only 0.21 and was not significant" : The

authors claim that SI and SIOD impacts are causing independently DPO. But I don't think so. Even if correlation is weak, years extracted for the composites for SI and SIOD may overlap each other. Please check the sample years used for making composite to verify whether enoughly different years are used for SI and SIOD for your discussion like with Figure4(c),(d) which are too similar.

13) L261 "(+)" : what does this represent?

14) Figure 5 (especially c,d ) panels are quite busy and hard to check the description in the texts. Please improve the visibility.

---

## Author Response (AR1)

**Reply to Referee 1:**

This manuscript discusses the dipole like pattern in summer time ozone concentrations (DP-$O_3$) seen in the north and south parts of China. The authors particularly focus on the roles of interannual variability (IAV) of climate dynamics like Rossby wave activity and propagation, and Indian ocean SST (dipole mode). Since $O_3$ pollution, being increasingly important, is now one of the biggest issues of air quality as well as $PM_{2.5}$ which is on the decrease in China, revealing underlying mechanisms of interannual changes and variability in surface $O_3$ including precursor emissions and meteorological factors is of importance. **I enjoyed much their discussions on the mechanisms that cause dipole anomaly patterns over China focusing on the roles of Arctic sea-ice and southern Indian ocean SSTs.**

1. However, the authors do seem to only focus on the aspect of climate dynamics, just looking at the meteorological parameters like pressure field, precipitation, and fail to address the actual mechanism of the dipole like $O_3$ anomalies over China. They do not seem to answer enoughly to the important questions of how meteorological changes give a rise surface $O_3$ anomalies in the region (upward/horizontal transport? Chemical loss? Chemical production?).

2. Furthermore, I do not fully agree with the analyzing approach adopted in this manuscript. For the $O_3$ variation, they use simulation with GEOS-CHEM, while for meteorological/climate parameters, another dataset like Era-5 is used instead. To keep consistency, the authors should have used the same meteorology as used in GEOS-CHEM.

I rate this manuscript as "major revision" at this moment, but think **this stuff could be published in ACP after additional analysis and modification are properly included**.

*Reply:*

(1) In our study, long-term changes in $O_3$ pollution were studied from the perspective of climate variability. In the revised version, we try our best to analyses the atmospheric chemical and physical processes related to changes of meteorological/climate parameters.

**Chemical and physical processes were examined using the outputs of GEOS-Chem**, in order to more clearly explore how meteorological changes lead to the distribution of DP-$O_3$. In the revised version, we have mainly discussed the effects of chemistry, planetary boundary layer (PBL) mixing, convention, transport and their sum. Accordingly, we **re-plotted new Figure 3d, Figure 4c and 4d** (Details are listed

in Reply to Comments M3).

(2) We must apologize that our negligence, using inconsistent datasets, made some confusions. In the revised version, all the meteorological data were **replaced with MERRA-2 data**, which did **not affect our conclusions**. Detailed comparisons are listed in Reply to Comments M1.

Major comments:

M1) The authors use different meteorological datasets for ozone (DP-O$_3$) and meteorological parameters (i.e. GEOS-Chem with MERRA-2, and ERA-5), isn't it possible to fix this inconsistency by synchronizing those two? Ideally, you should close the overall discussions within the framework of GEOS-Chem model simulation. At least, they should verify consistency between the Era-5 and MERRA2 in detail.

*Reply:*

(1) In the revised version, all the meteorological data were **replaced with MERRA-2 data.**

(2) The figures shown below compare results using MERRA-2 (left column) and ERA-5 (right column) data. The replacement of MERRA-2 data **did not affect our conclusions**.

**MERRA-2  ERA-5**

[Figure]

**New and old Figure 3 and 6.** The left column shows new figures using MERRA-2 data and the right column shows old figures using ERA-5 data.

*Revision:*

**p. 3, line 89:** The meteorological fields data with a horizontal resolution of 0.5° latitude by 0.625° longitude for the period 1980–2019 were taken from the **MERRA-2 dataset** (Gelaro et al., 2017), including geopotential height at 500 hPa (Z500) ......

M2) The DP-O$_3$ patterns are only evaluated with surface observations in China (Figure 1). But the DP-O$_3$ pattern could and should be further evaluated by using satellite observation like OMI/TROPOMI. This could make your later discussions (relationship with SI or SIOD) further clear and robust.

*Reply:*

The OMI/TROPOMI satellite observation were downloaded from https://https://disc.gsfc.nasa.gov/. After careful analysis, we found the satellite observation did not quite match our research topic. This study mainly focuses on the dominant pattern of **surface O$_3$ pollution** in summer in the east of China. OMI/TROPOMI satellite observations focus on **the total O$_3$ column or the tropospheric O$_3$**, which are different from the surface O$_3$ observations (Figure R1). The GEOS-Chem model used ozone concentrations at 10 meters above the surface so that the data from the **observational data could be reproduced** with the simulated data (Travis et al. 2019). So, the evaluation was only carried out through surface observations in the east of China.

[Figure]

**Figure R1.** (a) Spatial distributions of observed (unit: μg m$^{-3}$, dots) and OMI satellite

observed (unit: DU, shading) summer-mean MDA8 $O_3$ for the period 2015–2019.

*Related References:*

Travis K R and Jacob D J 2019 Systematic bias in evaluating chemical transport models with maximum daily 8 h average (MDA8) surface ozone for air quality applications: a case study with GEOS-Chem v9.02 Geophys. Model Dev. 12 3641–3648

M3) As pointed out above, you should more clearly discuss the actual mechanism of DP-$O_3$. At least, is should be separated into two categories: 1) transport (bot vertical and horizontal motion could affect surface $O_3$), and 2) chemistry (production and net-production of $O_3$ in the regions). You should evaluate the chemical roles by taking a look at chemical tendency (P and P-L) of $O_3$ in GEOS-Chem. For transport aspect, this can be evaluated by focusing on the concurrent changes in distribution of carbon monoxide (CO) or any other inert-like tracer. And if you look at CO, satellite observation (MOPITT etc.) may further help and reinforce the discussion on the transport aspects.

*Reply:*

Thank you for your professional advices. We have been trying our best to examine the outputs of our GEOS-Chem simulations, and we must say that we are not familiar with numerical simulation related to atmospheric chemistry. Although we decided to focus on climate dynamics at which we are skilled, we also added some primary discussions from the perspective of atmospheric chemical and physical processes. Accordingly, we re-plotted new Figure 3d, Figure 4 c and 4d.

**Chemical and physical processes were examined using the outputs of GEOS-Chem.** We used non-local PBL mixing in the simulation, so the emissions and dry deposition trends within the PBL were applied within the mixing (Holtslag and Boville, 1993). Compared with other terms, the value of wet deposition was extremely small, so it was not considered in this study (Liao et al., 2006). Consequently, the major chemical and physical processes related to meteorological conditions included the chemistry, convection, PBL mixing, transport and their sum within the PBL were the focus.

In order to provide a more quantitative evaluation of the contribution of chemical and physical processes, in Figure 3d, we examine the area-averaged differences in $O_3$ changes for NC and PRD. **Chemistry** represents the changes in net chemical

production, which appears to be the dominating process, leading to the greatest $O_3$ change between NC and the PRD (**12.3 Tons d$^{-1}$**, Figure 3d). Transport represents the change in horizontal and vertical advection of ozone. Depending on the ozone concentration gradient and wind anomalies, the **transport** difference between NC and PRD is **3.1 Tons d$^{-1}$** (Figure 3d). Convection changes slightly in NC and PRD. As the mixing process transports ozone along the vertical concentration gradient, it generally contributes negatively to the total ozone change. The above analysis indicates that different meteorological conditions between NC and the PRD led to the difference of $O_3$ concentration in the two regions (**differed by 5.2 Tons d$^{-1}$**), which eventually contributed the formation of DP-$O_3$.

The interannual variability in DP-$O_3$ is linked to the sea ice near the Franz Josef Land ($SI_{FJL}$) in May and the Subtropical Indian Ocean Dipole (SIOD) in JFM through large-scale circulation (i.e., $AC_{NC}$ and $C_{PRD}$). Large-scale anticyclonic (cyclonic) and cyclonic (anticyclonic) anomalies over NC and the PRD resulted in a sharp contrast of meteorological conditions between the above two regions. When $SI_{FJL}$ (SIOD) is positively abnormal, the **enhanced photochemistry in NC were 11.6 (12.3) Tons d$^{-1}$ stronger than that in the PRD** associated with $SI_{FJL}$ (SIOD) in Figure 4 cd, which plays an important role in the formation of the DP-$O_3$ pattern.

[Figure]

**Figure 3d.** Composites of the summer mass fluxes of $O_3$ associated with the DP-$O_3$ (DP-$O_3$P minus DP-$O_3$N) for the area-averaged differences (NC minus PRD) from 1980 to 2019. The bottom axis gives the names of the chemical and physical processes: chemical reaction (Chem), convection (Conv), PBL mixing (Mix), transport (Trans) and their sum (Sum).

[Figure]

**Figure 4.** Composite summer meteorological conditions, circulations and mass fluxes of $O_3$ associated with (c) $SI_{FJL}$ (positive $SI_{FJL}$ years minus negative $SI_{FJL}$ years) and (d) SIOD (positive SIOD years minus negative SIOD years) from 1980 to 2019. The bottom axis gives the names of the meteorological conditions and chemical and physical processes: the differences between $AC_{NC}$ and $C_{PRD}$ (unit: 10 gpm), surface incoming shortwave flux (Ssr, unit: W m$^{-2}$), surface air temperature (SAT, unit: K), and precipitation (Prec, unit: mm); chemical reaction (Chem, unit: Tons d$^{-1}$), convection (Conv, unit: Tons d$^{-1}$), PBL mixing (Mix, unit: Tons d$^{-1}$), transport (Trans, unit: Tons d$^{-1}$) and their sum (Sum, unit: Tons d$^{-1}$).

*Related References:*

Holtslag, A. and Boville, B. A.: Local versus nonlocal boundary layer diffusion in a global climate model, J. Climate, 6, 1825–1842, https://doi.org/10.1175/1520-0442(1993)006<1825:LVNBLD>2.0.CO;2, 1993.

Gong, C., Liao, H., Zhang, L., Yue, X., Dang, R., and Yang, Y.; Persistent Ozone Pollution Episodes in North China Exacerbated by Regional Transport, Environ. Pollut., 265, 115056, https://doi.org/10.1016/j.envpol.2020.115056, 2020.

Liao, H., Chen, W. T., and Seinfeld, J. H.: Role of climate change in global predictions of future tropospheric ozone and aerosols, J. Geophys. Res.-Atmos., 111, D12304, https://doi.org/10.1029/2005JD006852, 2006.

*Revision:*

**p. 4, line 114:** Chemical and physical processes were examined using the outputs of GEOS-Chem. Because non-local planetary boundary layer (PBL) mixing was used, emissions and dry deposition trends within the PBL were applied within the mixing (Holtslag and Boville, 1993). Compared with other terms, the value of wet deposition was extremely small, so it was not considered in this study (Liao et al., 2006). Consequently, the major chemical and physical processes related to meteorological conditions included the chemistry, convection, PBL mixing, transport and their sum within the PBL were the focus.

**p. 7, line 190:** In order to provide a more quantitative evaluation of the contribution of chemical and physical processes, in Figure 3d, we examine the area-averaged differences in $O_3$ changes for NC and PRD. **Chemistry** represents the changes in net chemical production, which appears to be the dominating process, leading to the greatest $O_3$ change between NC and the PRD (**12.3 Tons d$^{-1}$**, Figure 3d). Transport represents the change in horizontal and vertical advection of ozone. Depending on the ozone concentration gradient and wind anomalies, the **transport** difference between NC and PRD is **3.1 Tons d$^{-1}$** (Figure 3d). Convection changes slightly in NC and PRD. As the mixing process transports ozone along the vertical concentration gradient, it generally contributes negatively to the total ozone change. The above analysis indicates that different meteorological conditions between NC and the PRD led to the difference of $O_3$ concentration in the two regions (**differed by 5.2 Tons d$^{-1}$**), which eventually contributed the formation of DP-$O_3$.

**p. 10, line 226:** …… The chemical and physical processes of ozone production in GEOS-Chem simulations were analyzed. The difference of **chemical reactions** between NC and PRD had a large positive value (**11.6 Tons d$^{-1}$**), and the difference of the sum of all chemical and physical processes was 7.0 Tons d$^{-1}$ (Figure 4c), resulting in DP-$O_3$.

**p. 11, line 257:** …… The **chemical reactions increased 12.3 Tons d$^{-1}$** in NC comparing to those in the PRD (Figure 4 d), indicating that the strong solar radiation and high temperature conditions actually enhanced the chemical reactions in the atmosphere to produce more $O_3$ in NC.

M4) You need describe more about your GEOS-Chem simulations especially for the emissions. How does the model consider natural emissions like BVOCs and LNOx? In this manuscript, the authors state that they use GEOS-Chem simulations with fixed emissions. What kind of emission types are targeted for this treatment? (only anthropogenic or natural emissions also?) If the natural emissions are not fixed and follow the climate/meteorological variability, the authors should add description on how these natural emission IAVs affect DP-$O_3$. On the other hand, if the natural emissions are fixed as anthropogenic ones, you should in turn discuss how the natural emissions, if they are allowed to vary in response to climate condition, would cause additional effects.

*Reply:*

In our GEOS-Chem simulations, **both of the anthropogenic and natural emissions were fixed**, which means both of them do not have IAVS and also not varied in response to climate conditions.

(1) We **added this information into the manuscript to avoid ambiguity**.

(2) According to your suggestions, we found it is interesting to detect how the natural emissions, if they are allowed to vary in response to climate condition, would cause additional effects. Possibly, we will analyze it in our future work.

*Revision:*

**p. 6, line 143:** Based the above results, the GEOS-Chem model was then driven by **fixed anthropogenic and natural emissions** in 2010 and changing meteorological fields from 1980 to 2019 to highlight the impact of climate variability on $O_3$ concentration.

M5) For the dipole pattern over N/S China, the authors only try to attribute it to SI and SIOD. But in fact, this phenomenon should be also tightly linked to Asian monsoon and ENSO, shouldn't it? Please extend your discussions to cover the monsoon and ENSO influences.

*Reply:*

The interannual variations in strength of the EASM are commonly represented by the EASM index (EASMI). The EASMI introduced by Zhang et al. (2003) is used in this study. EASMI is defined as a difference of anomalous zonal wind between the (10°–20°N, 100°–150°E) and (25°–35°N, 100°–150°E) at 850 hPa during summer.

The composite $O_3$ concentration based on the EASMI indicates that the influence of EASM on $O_3$ is concentrated in the region **south of the Yellow River** (Figure R2). The **correlation coefficient between DP-O$_3$ and EASMI is –0.17**(insignificant) from 1980 to 2019.

There was **no significant** correlation between **DP-O$_3$ and Niño 3.4 index**, because the lead-lag correlation coefficient between DP-O$_3$ and Niño 3.4 index was maintained at a low level (Figure R3).

Although it is a simple linear relationship, we preliminarily believe that **ENSO**

**and EASM have no significant relationship with the DP-O₃.**

[Figure]

**Figure R2.** Composite differences of the detrended summer-mean MDA8 O₃ (unit: µg m⁻³) simulated by GEOS-Chem model between high and low EASMI years during 1980–2019. The white dots indicate that the composite differences are above the 90% confidence level.

[Figure]

**Figure R3.** Lead-lag correlation between the Niño 3.4 index and DP-O₃ time series. Negative (positive) lags indicate that the Niño 3.4 index is leading (lagging), and the horizontal dashed lines are the 0.05 significance levels.

*Related References:*

Zhang, Q. Y., Tao, S. Y., and Chen, L. T.: The inter-annual variability of East Asian summer monsoon indices and its association with the pattern of general circulation over East Asia (in Chinese). Acta Meteorologica Sinica, 61, 559–568, https://doi.org/10.11676/qxxb2003.056, 2003.

Minor comments:

1. L41: "were lower" --> "were lower by about *** ppbv"

*Reply:*

We have added this to the manuscript.

*Revision:*

**p. 2, line 41:** For example, the $O_3$ concentrations in the summers of 2012–2013 were lower by about **10 ppbv** than that in 2011 and 2014 (Chen et al. 2019).

2. L46: "enhancement of natural emissions of ozone precursors" Yes it's true, but it is not clear at all how natural emissions are treated in this study.

*Reply:*

This sentence is revised.

*Revision:*

**p. 2, line 45:** For example, Lu et al. (2019) **designed sensitivity simulations** to confirm that ozone pollution in China in 2017 was more serious than that in 2016, which was attributed to the large enhancement of nature emissions of ozone precursors caused by hot and dry climate condition in 2017.

3. L98-103: The math formulation should be further explained. What are f, z,U? and what do "_x", "_xy", and prime " ' " indicate? (I know what they are, but major part of readers can not catch them on)

*Reply:*

To make it easier to understand, we added the explanation of the math formulation in the manuscript.

*Revision:*

**p. 4, line 98:** The wave activity flux (WAF) was computed to illustrate the propagation of Rossby wave activities (Takaya and Nakamura 2001):

$$W = \frac{1}{2|\overline{U}|} \begin{bmatrix} \bar{u}(\psi'^2_x - \psi'\psi'_{xx}) + \bar{v}(\psi'_x\psi'_y - \psi'\psi'_{xy}) \\ \bar{u}(\psi'_x\psi'_y - \psi'\psi'_{xy}) + \bar{v}(\psi'^2_y - \psi'\psi'_{yy}) \end{bmatrix}$$

where **subscripts denote partial derivatives**; the overbar and prime represent

the climatological mean and anomaly, respectively; **$\psi'$ represents the stream function anomaly.** U is the horizontal wind speed; $u$ and $v$ are the zonal and meridional wind components, respectively; and $W$ denotes the two-dimensional Rossby WAF. The Rossby wave source $-\nabla \cdot V_\chi (f + \xi)$ proposed by Sardeshmukh and Hoskins (1988) is also calculated in this study. **V, $\xi$ and $f$ refer to the horizontal wind velocity, relative vorticity and geostrophic parameter**, respectively. **$\nabla$ is horizontal gradient; subscript $\chi$ represents divergent component**.

4. L136: Doesn't this version of CESM-LE consider chemistry? If so, you could discuss DP-O$_3$ in CESM-LE as well.

*Reply:*

The **CESM** Large Ensemble Project is a **global coupled climate model** simulations intended for advancing understanding of internal climate variability and climate change. **GEOS-Chem** is a **regional atmospheric chemical transport model**, which is widely applied to investigate the potential modulation of climatic variabilities on the anomalous distributions of pollutants (Li et al., 2019). In our study, CESM-LE was used to provide evidence of the influence of preceding climate variability on large-scale atmospheric circulations, rather than regional reaction process.

The version of CESM we used did not have enough members to output ozone concentrations, and the output **multilayer ozone concentrations were** somewhat **different from ground level O₃**. Therefore, we should not consider DP-O$_3$ discussion in CESM-LE.

*Related References:*

Li, K., Jacob, D. J., Hong, L., Zhu, J., Shah, V., Shen, L., Bates, K. H., Zhang, Q., and Zhai, S. X.: A two-pollutant strategy for improving ozone and particulate matter air quality in China, Nat. Geosci., 12, 906–910, https://doi.org/10.1038/s41561-019-0464-x, 2019.

5. L147-148: "the GEOS-Chem model has a good performance … Therefore EOF was applied": I don't understand the logic here. The sentences should be rephrased.

*Reply:*

This aforementioned sentence was revised.

*Revision:*

**p. 6, line 157:** As aforementioned, the GEOS-Chem model has a good performance in simulating surface $O_3$ concentration. The summer $O_3$ concentrations from 1980 to 2019 was simulated by GEOS-Chem, and the EOF approach was applied to the simulation data to explore the dominant patterns of summer mean $O_3$ pollution in the east of China.

6. L152: "the first EOF pattern …": What does this 1st EOF mode stand for as meteorological phenomenon?

*Reply:*

This study mainly discussed the existence of the north-south dipole pattern of the summer mean $O_3$ pollution in the east of China, and did not pay much attentions to the EOF1 of the monopole pattern.

The correlation coefficient between the time series of EOF1 and EASM index is +0.68 from 1980 to 2019. Yang et al. (2014) confirmed that $O_3$ concentration was high in central China in the strongest EASM years. Closely related to your former comments, we speculate that EOF1 may be related to the interannual variation of EASM.

*Related References:*

Yang, Y., Liao, H., and Li, J.: Impacts of the East Asian summer monsoon on interannual variations of summertime surface-layer ozone concentrations over China, Atmos. Chem. Phys., 14, 6867–6879, https://doi.org/10.5194/acp-14-6867-2014, 2014.

7. L176: "a moist, cool, … weak solar radiation were conductive to low $O_3$": Please check and discuss changes in chemical production and photochemical lifetime of $O_3$ in GEOS-Chem.

*Reply:*

Chemical and physical processes were examined using the outputs of GEOS-Chem. Chemistry represents the changes in **net chemical production**. The **Chemistry** associated with the moist, cool environment and weak solar radiation have large negative values (**−8.7 Tons d⁻¹**) in the PRD.

8. L192: ", but its effected" --> ", but its effects"

*Reply:*

The statement "but its effected" is revised to "but its effects".

*Revision:*

**p. 9, line 215:** Arctic SI in May was closely related to summer $O_3$ pollution in NC (Yin et al. 2019), but its **effects** on the north-south dipole distribution of $O_3$ had not been studied.

9. L216: "After removing the influences of ENSO": What are the ENSO influences like? And how did you remove them?

*Reply:*

(1). ENSO had no significant effect on $O_3$ dipole pattern (Figure R4). In order to explore the influence of $SI_{FJL}$ and SIOD independently on ENSO, ENSO influences were removed. There was no effect on the relationship between $SI_{FJL}$, SIOD and DP-$O_3$ after removing the influence of ENSO.

(2). The Niño 3.4 index is defined as the areal mean SST over the region covering 5°S–5°N and 170°–120°W. The method proposed by An (2003) is used to remove the signal of ENSO from the data of climate variables via removing the signal of Niño 3.4 index based on the following formula:

$$\xi = \xi^* - ENSO \times cov(\xi^*, ENSO) \, / \, var(ENSO)$$

where $\xi^*$ is the original time series of a climate variable (i.e., $SI_{FJL}$, SIOD), ENSO represents the time series of Niño 3.4 index, cov ($\xi^*$, *ENSO*) represents the temporal covariance between $\xi^*$ and ENSO, var (*ENSO*) represents the variance of ENSO, and $\xi$ represents the time series of this climate variable with the signal of ENSO removed.

[Figure]

**Figure R4.** Composites of (a) JFM and (b) May sea surface temperature associated with the DP-O$_3$ (i.e., DP-O$_3$P minus DP-O$_3$N) from 1980 to 2019. The green boxes in panels (a) and (b) mean the centers of the Nino3.4. The white dots indicate that the differences with shading was above the 90% confidence level.

*Related References:*

An, S. I. (2003). Conditional maximum covariance analysis and its application to the tropical Indian Ocean SST and surface wind stress anomalies. Journal of Climate, 16(17), 2932– 2938. https://doi.org/10.1175/1520-0442(2003)016<2932:CMCAAI>2.0.CO;2

10. L226: "82%" how did you draw this value?

*Reply:*

The SIOD has **11 years** significantly abnormal (i.e., |anomalies| > its one standard deviation, **bule bar in Figure R5**) from 1980 to 2019. In 11 years, the SIOD anomalies were homodromous with DP-O$_3$ for **9 years (blue dots in Figure R5)**. Therefore, when the SIOD anomalies were significant, the occurrence probability of DP-O$_3$ in the same phase is 82%.

[Figure]

**Figure R5.** The +SIOD (−SIOD) level indicates the SIOD index is larger (smaller) than the 1 (−1) × its standard deviation, while the +DP-O$_3$ (−DP-O$_3$) level indicates the index is positive (negative). The blue dots indicate the SIOD are with the same mathematical sign with the DP-O$_3$.

11. L227: "active centers" I didn't follow this.

*Reply:*

We have modified the contents of the manuscript to "**centers**".

*Revision:*

**p. 11, line 255:** Furthermore, the composite meteorological conditions in the positive and negative phases of SIOD had similar **centers** to that of DP-O$_3$.

12. L232: Note that the correlation coefficient between them was only 0.21 and was not significant": The authors claim that SI and SIOD impacts are causing independently DP-O$_3$. But I don't think so. Even if correlation is weak, years extracted for the composites for SI and SIOD may overlap each other. Please check the sample years used for making composite to verify whether enoughly different years are used for SI and SIOD for your discussion like with Figure4(c), (d) which are too similar.

*Reply:*

The SI$_{FJL}$ (SIOD) has 12(11) years significantly abnormal (i.e., |anomalies| > its one standard deviation) from 1980 to 2019. SI$_{FJL}$ and SIOD have few years of common significant anomalies (Figure R6), **more than 78% of the individual sample years were used to make composite with both indices**. We have improved this in our manuscript.

The **distribution of Figure 4(c), (d)** are given in the supplement (Figure S4,

S5), which are **actually quite different**. To facilitate comparison, Figure S4 and Figure S5 are redrawn as two columns. The **bar chart** is given in the manuscripts, in order to **save space and more clarity**.

[Figure]

**Figure R6**. Diagram of the number of anomalous years on $SI_{FJL}$ (red) and SIOD (blue) and common anomalies in both indices (purple) from 1980 to 2019.

[Figure]

**Figure S4 and S5.** Composite summer atmospheric circulations associated with the **SI_FJL (left)** and **SIOD (right)** for the period 1980 to 2019, including (a, d) surface air temperature (SAT, unit: K, shadings) and geopotential height at 500 hPa (unit: 10 gpm, contours), (b, e) surface incoming shortwave flux (Ssr, unit: W m$^{-2}$, shadings) and low and medium cloud cover (Mlcc, unit: 1, contours), and (c, f) precipitation (Prec, unit: mm, shadings) and surface wind (unit: m s$^{-1}$, arrows). The white dots indicate that the composites with shading were above the 90% confidence level. The black boxes in (a) and (d) indicate the centers of the AC$_{NC}$ and C$_{PRD}$, respectively. The green boxes in (b), (c), (e) and (f) represent the areas of NC and the PRD.

*Revision:*

**p. 11, line 261:** Changes in $SI_{FJL}$ and SIOD both could possibly contribute to the formation of DP-$O_3$. Note that $SI_{FJL}$ and SIOD have few years of common significant anomalies, **more than 78% of the individual sample years were used to make composite with both indices**. The correlation coefficient between them was only 0.21 and was not significant ……

13. L261 "(+)" : what does this represent?

*Reply:*

The "(+)" means the **positive anomalies of geopotential height**, that is, the anomalous anticyclonic circulation.

14. Figure 5 (especially c,d ) panels are quite busy and hard to check the description in the texts. Please improve the visibility.

*Reply:*

We are so sorry for the difficulty in your reading, and **Figure 5 was replotted to show the information in a clearer way.**

*Revision:*

**p. 12, line 279:**

[Figure]

**Figure 5.** Composites of (a) May Arctic SST (unit: K), (c) velocity potential (unit: $10^5$ $m^2\ s^{-1}$, shading) and divergent wind (unit: m s$^{-1}$, arrows), and (e) Rossby wave source anomalies at 500 hPa (unit: $10^{-11}\ s^{-2}$) associated with $SI_{FJL}$ index (negative $SI_{FJL}$ years minus positive $SI_{FJL}$ years) from 1980 to 2019. The back box in (a) and (b), yellow box in (c) and (e) and green box in (d) and (f) represents the center of the SST, velocity potential and Rossby wave source anomaly associated with $SI_{FJL}$, respectively. The white dots indicate that the composites with shading were above the 90% confidence level.

**Reply to Referee 2:**

The manuscript compares a relatively short (2015 – 2019) observational record of June-July-August average ozone over eastern China to output from the GEOS-Chem model. With the ability of GEOS-Chem to adequately simulate ozone demonstrated, the authors analyze an extended GEOS-Chem simulation over 1980 – 2019 driven by MERRA-2 but using constant anthropogenic emissions. Through EOF analysis, they find a dipole in ozone variability (DP-O$_3$), with centers of opposite sign over North China (NC) and the Pearl River Delta (PRD). The interannual variability in DP-O$_3$ is then linked to variations in May sea-ice extent in the Arctic ocean to the north of Svalbard and Franz Josef Land (SI$_{FJL}$). Variability in sea-ice gives rise to anomalous Rossby wave forcing that propagates downstream and produces meteorological conditions over China that modify the photochemical production and accumulation of ozone. The authors similarly make a link between DP-O$_3$ and variability in sea surface temperatures over the Indian Ocean, the Subtropical Indian Ocean Dipole (SIOD). These links between SI$_{FJL}$ and SIOD and meteorological conditions over China are then reinforced through analysis of the variability in a large ensemble (40 members) of the CESM model. To close, the SI$_{FJL}$ and SIOD indices are combined to create a new index (SEI) that explains a larger percentage (~40%, r=0.62) of the variance in DP-O$_3$.

**The paper is generally well organized and the development of the statistical relationships are easy to follow.** I do not have a strong background in statistical analysis but the use of correlation and composites seems solid, particularly for derived impacts of sea-ice variability which has parallels with similar hypothesized links between sea-ice and interannual variability of the east Asian monsoon. The effects of the Subtropical Indian Ocean Dipole SST anomalies in January-February-March are statistically weaker and the authors suggest a mechanism involving subsurface heat content anomalies that migrate across the Indian ocean towards Sumatra by June-July-August that seems speculative. Overall, though, **I only have minor comments that are itemized here.**

1. Lines 89 – 90: It might be helpful for the reader if the mention of the fifth generation ECMWF reanalysis also included the phrase 'ERA5', as it is frequently referred to.

*Reply:*

According to the advice of the other reviewer, the data has been replaced with **MERRA-2 data** to enhance the consistency of data.

**This phrase, i.e., MERRA-2, has been used** in the manuscript.

*Revision:*

**p. 3, line 89:** The meteorological fields data with a horizontal resolution of

0.5° latitude by 0.625° longitude for the period 1980–2019 were taken from the **MERRA-2 dataset** (Gelaro et al., 2017), including geopotential height at 500 hPa (Z500) ......

2. Lines 123 – 126: The spatial correlation coefficient is definitely a significant aspect of assessing model performance, but since the paper is about the dipole pattern the temporal behavior will be important. Can the authors present one or two widely used metrics, such as root mean square error of MDA8 ozone for the 2015 – 2019 period? Preferably, this would be presented separately for both the NC and PRD regions.

*Reply:*

In the revised version, two widely used indicators, **root mean square error / mean and mean absolute error (MAE)**, were used to evaluate the simulation performance of GEOS-Chem model.

Compared the simulated and observed summer mean MDA8 $O_3$ concentrations in NC and the PRD, which had a low bias with a **MAE** of **5.7 μg m$^{-3}$ and 12.1 μg m$^{-3}$** in the PRD and NC, respectively. The values of **root mean square error / mean** were **15.8 % and 8.1 %** in NC and the PRD, respectively, indicating the good performance of reproducing the $O_3$ concentration.

*Revision:*

**p. 5, line 132:** …… Compared the simulated and observed summer mean MDA8 $O_3$ concentrations in NC and the PRD, which had a low bias with a mean absolute error of **5.7 μg m$^{-3}$ and 12.1 μg m$^{-3}$** in the PRD and NC, respectively. The values of root mean square error / mean were **15.8 % and 8.1 %** in NC and the PRD, respectively ……

3. Line 143: '…three years with the lowest and highest simulated SI in 143 each member.' Was sea-ice for a particular region, such as the Barents sea mentioned earlier, used or was it overall Arctic sea-ice extent? Was a particular month used?

*Reply:*

Based on the conclusions in the manuscript, the three years with the lowest and highest **SI anomalies near the Franz Josef Land in May** and **SIOD anomalies in January–February–March** were selected. We modified the manuscript to avoid

ambiguity.

*Revision:*

**p. 6, line 153:** …… composite analyses were conducted based on the three years with the lowest and highest simulated **preceding climatic variability for a particular month** in each member.

4. Line 168: Just to clarify, the timeseries of the DP-$O_3$ timeseries shown in panel (a) of Figure 2 is the JJA average of the EOF each year? But the EOF is calculated daily?

*Reply:*

EOF is calculated using an **annual summer average**. The EOF approach was applied to summer-mean $O_3$ for the period 1980–2019.

*Revision:*

**p. 7, line 179:** The time series of DP-$O_3$ showed a strong interannual variation ……

5. Lines 184 – 185: The caption on Figure 2 should have some brief description of what SI$_{FJL}$, SIOD and SEI are. I know these are all described in detail a bit further down in the text, but a brief description to give the reader some idea of what is being presented is helpful.

*Reply:*

In the revised version, we have added some brief description of SI$_{FJL}$, SIOD and SEI in the caption on Figure 2.

*Revision:*

**p. 8, line 201:** Figure 2. Variations in standardized DP-$O_3$ time series (black), **May SI near the Franz Josef Land** (SI$_{FJL}$, red), **January–February–March mean Subtropical Indian Ocean Dipole** (SIOD, blue), and synergistic effects index (SEI, green) from 1980 to 2019. **SEI defined as the weighted average of SI$_{FJL}$ and SIOD**. The correlation coefficients of the DP-$O_3$ with SI$_{FJL}$ (red), SIOD (blue), and SEI (green) were shown in the figure.

6. Lines 187 – 191: Some of the acronyms used in the caption for Figure 3 are easily

understood, but others like Ssr and MIcc are not.

*Reply:*

Some explanations have been **added to the acronyms** used in the caption for Figure 3 to make it easier to understand.

*Revision:*

**p. 9, line 206:** Figure 3. Composite summer atmospheric circulations associated with the DP-O$_3$ (DP-O$_3$P minus DP-O$_3$N) for the period 1980 to 2019, including (a) **surface air temperature** (**SAT**, unit: K, shadings) and geopotential height at 500 hPa (unit: gpm, contours), (b) **surface incoming shortwave flux** (**Ssr**, unit: W m$^{-2}$, shadings) and **low and medium cloud cover** (**Mlcc**, unit: 1, contours), and (c) **precipitation** (**Prec**, unit: mm, shadings) and surface wind (unit: m s$^{-1}$, arrows) ……

7. Lines 195 – 196: Was there any detrending of the sea-ice anomalies performed before the calculation of SI$_{FJL}$?

*Reply:*

We removed the trend before calculating all indices and, to be sure, the trend of sea ice anomalies is also **removed** before the calculation of SI$_{FJL}$.

8. Line 249: The presentation of May SST anomalies in panel (a) of Figure 5 show positive anomalies extending well north of 80N. Isn't there almost always sea-ice present in this area?

*Reply:*

Thank you for pointing out our mistake that the missing value was not properly handled when using SST data. The improper control of data quality was revised throughout the manuscript. SST in the Barents and Kara Sea (black box in Figure 5) is used to calculate sea-ice related SST indices. Therefore, the **negligence does not affect our conclusions** because we did **not use SST data north of 80°N**.

*Revision:*

**p. 12, line 279:**

[Figure]

**Figure 5.** Composites of (a) May Arctic SST (unit: K) associated with $SI_{FJL}$ index (negative $SI_{FJL}$ years minus positive $SI_{FJL}$ years) from 1980 to 2019. The back box in (a) and (b) represents the center of the SST associated with $SI_{FJL}$. The white dots indicate that the composites with shading were above the 90% confidence level.

9. Lines 272 – 273: Was there a range of years from which the three highest and lowest $SI_{FJL}$ anomalies were taken from the CESM large ensemble?

*Reply:*

    **The period 1980–2019** was used to obtain three years with the highest and lowest $SI_{FJL}$ anomalies. To make this clear, we added a range of years.

*Revision:*

    **p. 13, line 302:** The relationship between the preceding May SI anomalies and the JJA EU-like pattern was also confirmed by large ensemble simulations of CESM **during 1980–2019**.